# *Toxoplasma gondii* PPM3C, a secreted protein phosphatase, affects parasitophorous vacuole effector export

Joshua Mayoral[1], Tadakimi Tomita[1], Vincent Tu[1], Jennifer T. Aguilan[2], Simone Sidoli[2], Louis M. Weiss[1,3]*

**1** Department of Pathology, Albert Einstein College of Medicine, Bronx, New York, United States of America, **2** Department of Biochemistry, Albert Einstein College of Medicine, Bronx, New York, United States of America, **3** Department of Medicine, Albert Einstein College of Medicine, Bronx, New York, United States of America

* louis.weiss@einsteinmed.org

**Data Availability Statement:** Proteomic data has been deposited into the mass spectrometry open access repository Chorus under Project ID 1695.

## Abstract

The intracellular parasite *Toxoplasma gondii* infects a large proportion of humans worldwide and can cause adverse complications in the settings of immune-compromise and pregnancy. *T. gondii* thrives within many different cell types due in part to its residence within a specialized and heavily modified compartment in which the parasite divides, termed the parasitophorous vacuole. Within this vacuole, numerous proteins optimize intracellular survival following their secretion by the parasite. We investigated the contribution of one of these proteins, TgPPM3C, predicted to contain a PP2C-class serine/threonine phosphatase domain and previously shown to interact with the protein MYR1, an essential component of a putative vacuolar translocon that mediates effector export into the host cell. Parasites lacking the TgPPM3C gene exhibit a minor growth defect *in vitro*, are avirulent during acute infection in mice, and form fewer cysts in mouse brain during chronic infection. Phosphoproteomic assessment of TgPPM3C deleted parasite cultures demonstrated alterations in the phosphorylation status of many secreted vacuolar proteins including two exported effector proteins, GRA16 and GRA28, as well as MYR1. Parasites lacking TgPPM3C are defective in GRA16 and GRA28 export, but not in the export of other MYR1-dependant effectors. Phosphomimetic mutation of two GRA16 serine residues results in export defects, suggesting that de-phosphorylation is a critical step in the process of GRA16 export. These findings provide another example of the emerging role of phosphatases in regulating the complex environment of the *T. gondii* parasitophorous vacuole and influencing the export of specific effector proteins from the vacuolar lumen into the host cell.

## Author summary

The flexible life cycle of the intracellular parasite *Toxoplasma gondii* allows it to infect many different types of warm-blooded hosts, as well as diverse cell types once inside the host organism. This formidable achievement is partly mediated by the establishment of a

The proteomic data was also deposited in ToxoDB (EuPathDb).

**Funding:** R01AI134753 (LMW) and F31AI136401 (J.M.) National Institutes of Health/National Institute of Allergy and Infectious Diseases https:// www.niaid.nih.gov/ T32GM007288 (JM) National Institutes of Health/National Institute of General Medical Sciences https://www.nigms.nih.gov/. This work was also supported by P30CA013330, SIG #1S10OD016214-01A1, and SIG #1S10OD019961-01 (Einstein Analytical Imaging Facility) The funders had no role in study design, data collection and analysis, decision to publish, or preparation of the manuscript.

**Competing interests:** The authors have declared that no competing interests exist.

unique compartment following host cell invasion, termed the parasitophorous vacuole. While advancements have been made in cataloguing *Toxoplasma* secreted proteins that reside within this vacuole, the specific functions and contributions of many of these secreted parasite "tools" remain elusive. Here, we assessed the contribution of a parasite vacuolar protein called TgPPM3C, predicted to function as an enzyme that dephosphorylates other proteins. We found that deleting the TgPPM3C gene in the parasite results in a profound virulence defect during infection in mice, likely due to the dysregulated phosphorylation status of many vacuolar proteins detected by phosphoproteomic analysis of TgPPM3C-deleted parasites. We found that the phosphorylation status of one such protein, GRA16, influences its ability to cross the parasitophorous vacuole membrane and enter the host cell, where it is known to induce host transcriptional changes that benefit parasite growth. These findings illustrate the emerging role of *Toxoplasma* vacuolar phosphatases in regulating host-parasite interactions during infection.

## Introduction

A large proportion of the human population, varying by region, is predicted to be infected with the parasite *Toxoplasma gondii* based on seropositivity, posing a life-threatening risk in immune-compromised individuals [1]. *T. gondii* infects not only an impressive range of warm-blooded hosts, but can also infect any nucleated cell type within the host [2]. During acute infection, the rapidly growing tachyzoite life stage disseminates to various host tissues through repeated rounds of invasion, replication, and host cell lysis [3]. Following an adequate host immune response, the number of tachyzoites in the host declines, and a subset of tachyzoites differentiate into a quasi-dormant life stage termed the bradyzoite [4]. Bradyzoites, packed within large intracellular tissue cysts, are found predominately in the brain and muscle tissue and are a hallmark of chronic infection, persisting for an indefinite period of time in both rodents and humans [5]. Although advances have been made in understanding how the parasite thrives within host cells and avoids host cell defenses, much remains to be discovered regarding how the unique compartment in which parasites replicate, the parasitophorous vacuole (containing tachyzoites) or the tissue cyst (containing bradyzoites), is optimized and remodeled throughout the infection process of either life stage.

During intracellular development in both tachyzoite and bradyzoite life stages, a wide array of proteins are secreted into the vacuolar compartment, many of which originate from dense granule organelles (referred to as "GRA" proteins) and potentially other secretory vesicles within the parasite [6]. GRA proteins mediate diverse roles during parasitic development after their secretion into the vacuole: for example, GRA2 and GRA6 are pivotal in forming the distinct membranous tubules of the intravacuolar network (IVN) [7], while GRA17 and GRA23 mediate the passive transport of small molecules across the parasitophorous vacuole membrane [8]. Some GRA proteins have been shown to operate outside of the parasitophorous vacuole lumen, influencing host cell signaling not only from the parasitophorous vacuole membrane, but also within the host cell cytoplasm and nucleus following their export from the vacuole [9,10].

Despite the progress made in understanding the function of certain GRA proteins, much less is known regarding the functional consequences of post-translational modifications found in many of these same GRA and vacuolar proteins. Numerous vacuolar proteins have been shown to be phosphorylated [11], and evidence exists for their phosphorylation post-secretion into the vacuole [11,12]. The vacuolar kinase WNG1 was recently shown to be pivotal in

organizing the IVN within the parasitophorous vacuole through the phosphorylation of GRA proteins, in turn affecting the affinity of GRA4, GRA6, and GRA7 to lipid membranes [13]. The putative kinases ROP21 and ROP27 have been shown to localize to the parasitophorous vacuole lumen and cyst matrix of tachyzoite vacuoles and *in vitro* tissue cysts respectively, but not to rhoptry organelles as their initial classification implied [14]. Simultaneous deletion of ROP21 and ROP27 results in reduced cyst burdens in mouse brain, suggesting that their catalytic activity within the cyst matrix is pivotal to the establishment or maintenance of cysts during chronic infection [14]. Measures to counterbalance vacuolar protein phosphorylation are likely provided by putative vacuolar phosphatases [15], although little is known regarding the outcomes of protein de-phosphorylation within the parasitophorous vacuole or tissue cyst.

Towards elucidating the molecular basis behind chronic *Toxoplasma* infection, one intensively studied structure specific to the bradyzoite life stage is the cyst wall, appearing as a dense collection of filamentous and vesicular structures underneath a delimiting cyst membrane [16]. Many proteins secreted by replicating parasites have been shown to accumulate at the cyst wall, several of which are uniquely expressed in the bradyzoite life-stage [17–19]. Our research group has identified a set of proteins that are enriched in an *in vitro* derived cyst wall fraction obtained from parasites grown under alkaline-stress conditions [20]. Many of these proteins demonstrate no homology to known proteins of other organisms and are expressed by both tachyzoite and *in vitro* bradyzoite life stages, suggesting that these proteins serve functions in both tachyzoite vacuoles and bradyzoite tissue cysts. In the current study, we have characterized one of these putative cyst wall proteins, previously dubbed TgPPM3C [15], which is predicted to contain a recognizable domain belonging to the PP2C-class of serine/threonine phosphatases. We hypothesized that TgPPM3C could influence the viability of parasites by affecting the phosphorylation status of vacuolar protein substrates.

We tested our hypothesis by first determining the localization of epitope-tagged TgPPM3C in tachyzoite and bradyzoite vacuoles and proceeded with ablating TgPPM3C protein expression. In TgPPM3C knockout parasites (ΔTgPPM3C) a minor growth defect was observed in human fibroblast cultures *in vitro*, a complete loss of virulence was observed during acute infection in mice, and a decrease in brain cyst burden during chronic infection. Both the virulence loss and brain cyst burden phenotypes were rescued by complementation of the TgPPM3C gene. The phosphoproteome of ΔTgPPM3C cultures was compared to cultures of parasites expressing TgPPM3C and this demonstrated that phosphopeptides from several vacuolar proteins were significantly more abundant in ΔTgPPM3C cultures, potentially revealing TgPPM3C substrates. Based on clues offered by phosphoproteomic and co-immunoprecipitation results, we next assessed whether protein effector export from the parasitophorous vacuole into the host cell was altered in the ΔTgPPM3C strain. This revealed that the export of two protein effectors, GRA16 and GRA28, were impaired in the ΔTgPPM3C strain, though the export of other protein effectors were seemingly not compromised. Mutations mimicking the phosphorylation of two GRA16 serine residues also results in GRA16 export defects in wild type parasites, though importantly not in ΔTgPPM3C parasites. Overall, these results demonstrate that TgPPM3C influences the phosphorylation status of many vacuolar proteins and reveal de-phosphorylation as a potentially critical process that facilitates the export of GRA16 from the parasitophorous vacuole.

## Results

### TgPPM3C is a putative protein phosphatase secreted into the lumen of the parasitophorous vacuole

The TgPPM3C gene (Gene ID: TGME49_270320) is predicted to encode a protein with a signal peptide and a catalytic PP2C domain (Fig 1A). Compared to human PPM1A, TgPPM3C

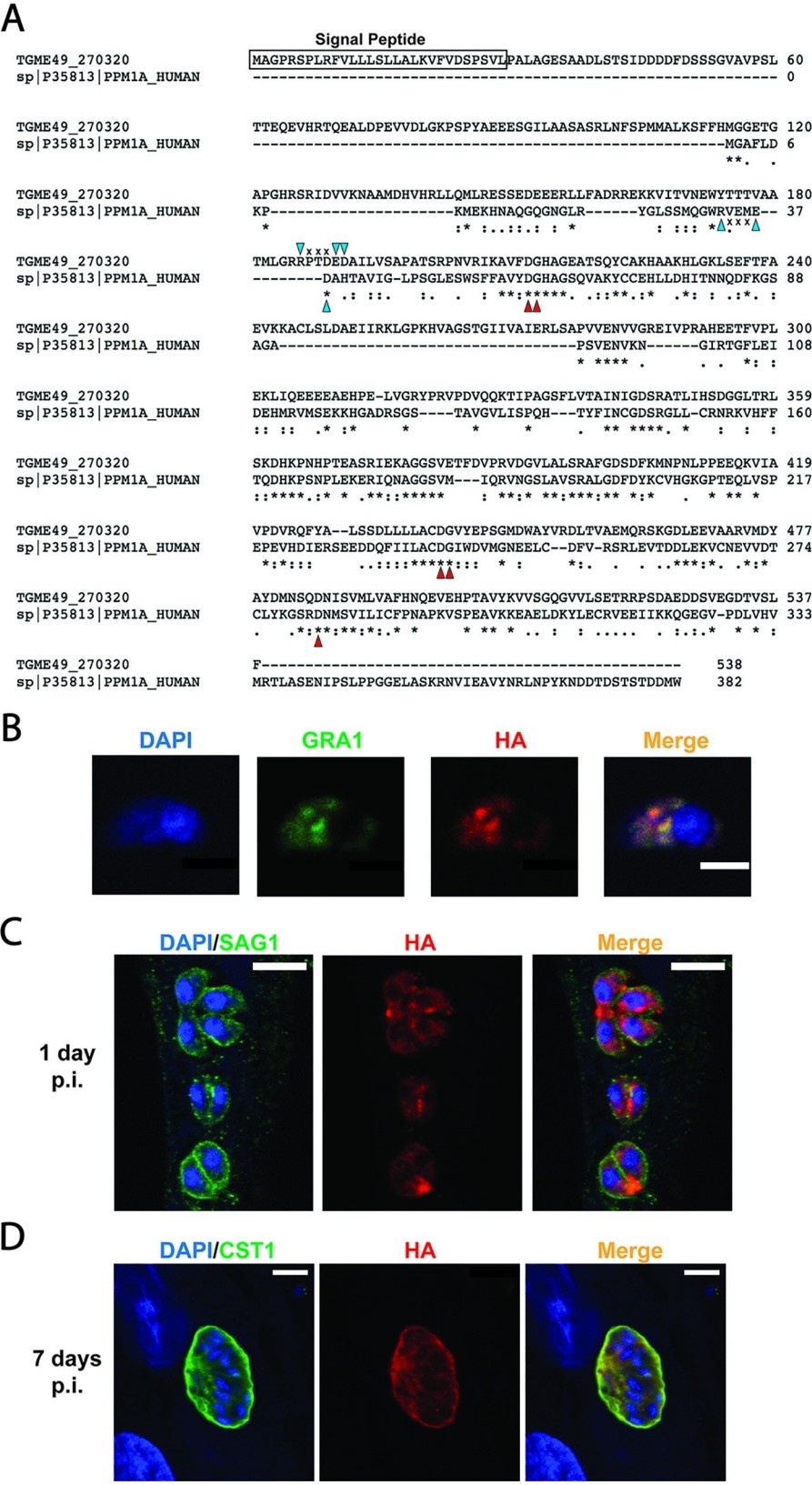

**Fig 1. TgPPM3C is a parasitophorous vacuole protein with a PP2C-class phosphatase catalytic domain. (A)** Amino acid sequence alignment of TgPPM3C (Gene ID: TGME49_270320) and human protein phosphatase 1a

(PPM1a), a canonical PP2C class phosphatase, performed with Clustal Omega [58]. TgPPM3C contains a predicted signal peptide (boxed region) and an extended N-terminal domain upstream of the conserved PP2C catalytic domain. Residues important for metal binding are indicated with red arrows. Unaligned but conserved residues important for phosphate and metal binding shared by TgPPM3C and PPM1A are indicated by blue arrows. **(B)** Immunofluorescence image of an extracellular parasite with labeled dense granules (GRA1), endogenously tagged TgPPM3C-HA (HA), and nucleus (DAPI). Partial colocalization is observed between GRA1 and HA, indicating TgPPM3C-HA is potentially packaged into dense granules. Scale bar equals 5μm. **(C, D)** Immunofluorescence image of parasitophorous vacuoles grown under tachyzoite growth conditions (24 hours post-infection) or bradyzoite growth conditions (**D,** 7 days post-infection). Parasites in **(C)** are labeled with SAG1, while bradyzoite differentiation in **(D)** was probed by glycosylated CST1 expression, detected with SalmonE monoclonal antibody. Vacuolar TgPPM3C is observed more prominently after prolonged culture and parasite development (as in **D**). Scale bars indicate 10μm.

contains an extended N-terminal region relative to the catalytic domain, but still shares conserved amino acid residues involved in phosphate and $Mn^{2+}/Mg^{2+}$ binding (Fig 1A, blue and red arrows). To identify the localization of TgPPM3C, the endogenous locus of the gene was epitope-tagged using Cas9 guide RNA targeting the C-terminus of TgPPM3C. A single-copy of the hemagglutinin epitope (HA) was fused to the C-terminus in the Prugniaud Δku80Δhxgprt (PruQ) background. TgPPM3C-HA was found to partially co-localize with GRA1 in extracellular tachyzoites (Fig 1B), indicating TgPPM3C is potentially secreted through the dense granule secretory pathway. Following 1 day of infection under tachyzoite growth conditions, TgPPM3C-HA was demonstrated to localize predominately within developing parasites, although some signal can also be seen within the lumen of the parasitophorous vacuole (Fig 1C). This localization within the lumen is in agreement with, though less apparent, than the previously described vacuolar localization of endogenously tagged TgPPM3C-3xHA in RH strain parasites [21]. Bradyzoite induction via growth in alkaline serum-depleted media demonstrates a more pronounced accumulation of secreted TgPPM3C within *in vitro* tissue cysts 7 days post-infection (Fig 1D), suggesting TgPPM3C may also operate within the lumen of tissue cysts during parasite differentiation.

## ΔTgPPM3C parasites exhibit a growth defect *in vitro* and a profound virulence defect *in vivo*

To begin assessing the role of TgPPM3C, expression of TgPPM3C-HA protein was removed by deleting the predicted start codon of TgPPM3C and introducing a multi-stop codon sequence. Using the inserted multi-stop codon sequence as a handle for Cas9 guide RNA targeting, TgPPM3C protein was then restored in the ΔTgPPM3C strain to yield a complemented strain (TgPPM3C-COMP). Immunoblotting of tachyzoite lysates demonstrates TgPPM3C-HA migration close to the predicted size of the protein (59kDa, Fig 2A). As expected, TgPPM3C-HA protein is undetectable in the ΔTgPPM3C strain, and comparable amounts of TgPPM3C protein are present in the TgPPM3C-COMP strain compared to the parental TgPPM3C-HA strain (Fig 2A).

To assess for growth defects in ΔTgPPM3C parasites, plaque assays of human foreskin fibroblast (HFF) monolayers were performed, allowing the TgPPM3C-HA, ΔTgPPM3C, and TgPPM3C-COMP strains to replicate for two weeks. The sizes of plaques generated from multiple rounds of the tachyzoite lytic cycle were measured and compared between each strain. A subtle but significant decrease in average plaque size was noted in the ΔTgPPM3C strain compared to the TgPPM3C-HA and TgPPM3C-COMP strains (Fig 2B). *In vivo* virulence was next assessed following intraperitoneal infection of C57Bl/6 mice with 16,000 parasites per mouse. Although more than half of the mice infected with TgPPM3C-HA and TgPPM3C-COMP strains succumbed to infection, no mice infected with ΔTgPPM3C parasites perished, revealing a profound defect in virulence (Fig 2C). Cyst burden was measured from the brains of Bl/6

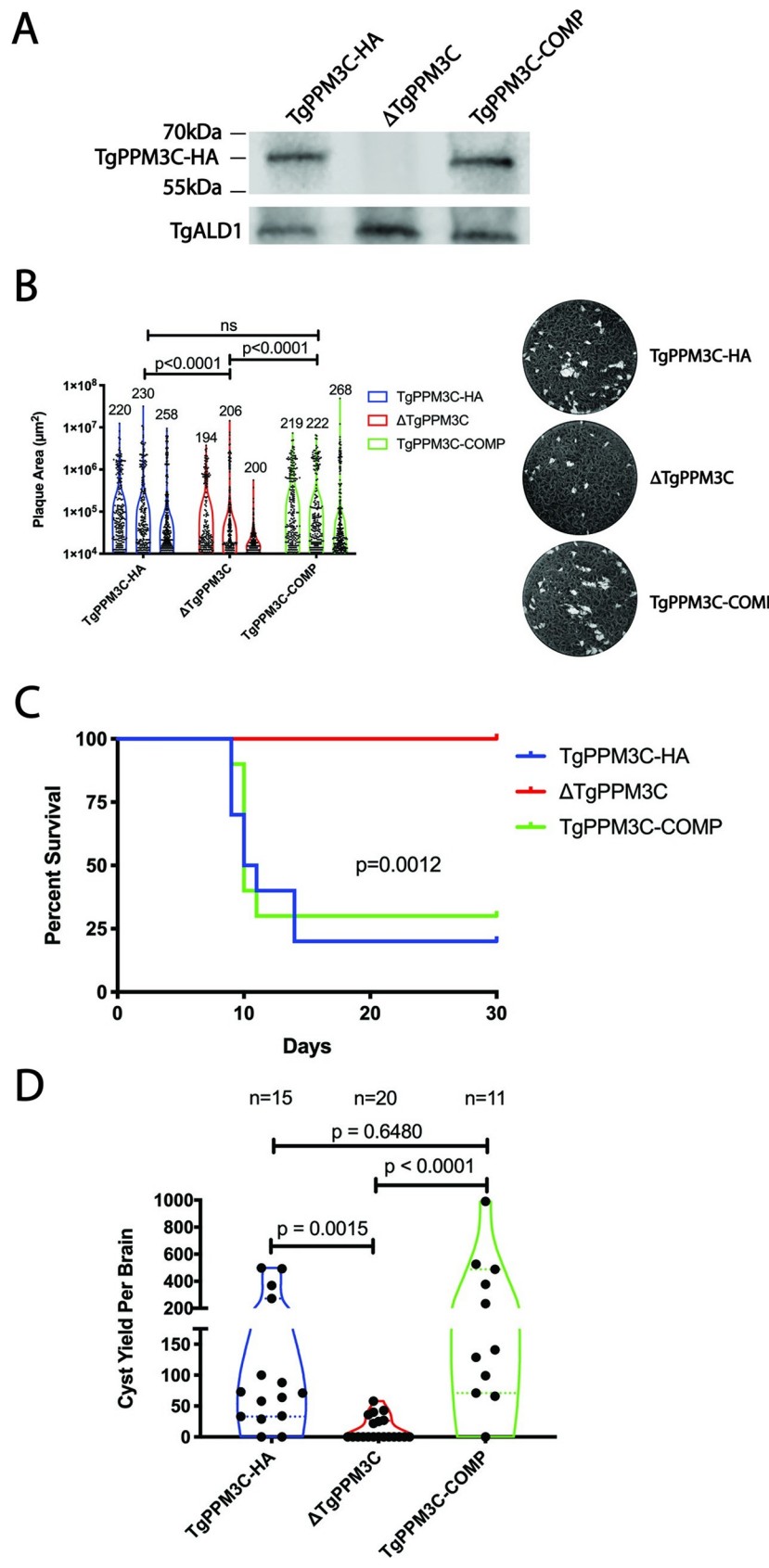

**Fig 2. ΔTgPPM3C parasites exhibit growth defects *in vitro* and *in vivo*. (A)** Immunoblot images obtained from SDS-PAGE separated protein lysates of tachyzoite infected cultures, 24 hours post-infection. Comparable amounts of TgPPM3C protein, migrating close to the predicted size (59kDa), are expressed in TgPPM3C-HA and TgPPM3C-COMP infected cultures but not in ΔTgPPM3C infected cultures. TgALD1 was used as a parasite specific loading control. **(B)** Violin plots depicting the distribution of plaque sizes formed by TgPPM3C modified strains after two weeks of growth in human fibroblast monolayers. Representative images of wells containing plaques from each strain are shown to the right. Violin plots represent three independent experiments for each strain. The number of plaques formed by each strain across multiple wells for a given experiment are indicated above each violin plot. Plaque size was quantified in ImageJ, using the Kruskal-Wallis and Dunn's multiple comparisons test to compare each group and generate p values. A significant decrease in plaque size is observed in the ΔTgPPM3C strain compared to TgPPM3C-HA and TgPPM3C-COMP strains. **(C)** Kaplan-Meier survival curves from C57Bl/6 mice injected intraperitoneally with equal amounts of parasites (16,000) from each strain. 10 mice were injected per group. 20% and 30% of mice survived after 30 days following infection with TgPPM3C-HA and TgPPM3C-COMP parasites respectively, whereas all mice survived following infection with ΔTgPPM3C parasites, indicating ΔTgPPM3C parasites are attenuated. A log-rank test indicates a significant difference in survival curves between groups. Data are representative of two independent experiments. **(D)** Violin plots of cyst burden from individual C57Bl/6 mouse brains, harvested 30 days post-infection following intraperitoneal injection with 2,000 parasites of each strain. Cyst burden is estimated based on cyst counts from one cerebral hemisphere per mouse brain. The number of brains from which cysts were quantified are indicated above each plot. Significantly less cysts are formed by ΔTgPPM3C parasites compared to TgPPM3C-HA and TgPPM3C-COMP parasites. Data are pooled from two independent experiments.

mice chronically infected with each strain 30 days after intraperitoneal infection with 2,000 parasites. A significant reduction in ΔTgPPM3C cyst burden was measured compared to the TgPPM3C-HA and TgPPM3C-COMP strains (Fig 2D). No significant differences in virulence or cyst burden were observed between the TgPPM3C-HA and TgPPM3C-COMP strains (Fig 2D).

To determine whether the growth defect observed in the ΔTgPPM3C strain during *in vitro* growth in fibroblasts could be due to defects in either invasion, parasite replication, or host cell egress, we probed for each of these events in the parental TgPPM3C-HA, ΔTgPPM3C, and TgPPM3C-COMP strains (S1 Fig). There were no significant differences in the knockout strain compared to parental and complemented strains when counting intracellular vs. extracellular parasites 30 minutes post-infection (invasion assay), when counting the number of parasites per vacuole 32 hours post-infection (replication assay), or when counting occupied vs. unoccupied vacuoles following DMSO or A23178 calcium ionophore treatment (egress assay). Thus, the growth defect detected by plaque assay in ΔTgPPM3C parasites might be due to subtle differences in growth during the course of intracellular infection that were undetected when assessing various aspects of parasite growth utilizing these standard assays.

## Co-immunoprecipitation of TgPPM3C-HA enriches only a few parasite proteins

Based on the PP2C domain of TgPPM3C, we hypothesized that secreted vacuolar proteins normally dephosphorylated by TgPPM3C remain phosphorylated in ΔTgPPM3C parasites. As a consequence, the loss of phospho-regulation of one or several key vacuolar proteins results in the virulence and growth defects observed in the ΔTgPPM3C strain. Tachyzoite vacuoles from TgPPM3C-HA and ΔTgPPM3C strains examined by transmission electron microscopy demonstrating that there was no difference in gross morphology between these strains (S2 Fig), indicating that an obvious disruption of parasitophorous vacuole architecture was not responsible for the ΔTgPPM3C phenotype. We next sought to identify TgPPM3C interacting parasite proteins, performing co-immunoprecipitation experiments from tachyzoite infected cultures 24 hours post-infection using anti-HA magnetic beads to pulldown TgPPM3C-HA protein. As a negative control, Co-IPs of non-epitope tagged PruQ parasites with anti-HA magnetic beads were also performed. Based on two independent experiments, apart from

TgPPM3C-HA, we identified two GRA proteins (GRA7, GRA9) and two parasite proteins known to be secreted into the parasitophorous vacuole (MAG1, MYR1) as significantly enriched in both experiments (p < 0.05, log$_2$ fold-change > 1.5) (Table 1, S1 Data). Intriguingly, MYR1 is known to be essential to the process of parasite protein export from within the parasitophorous vacuole into the host cell across the parasitophorous vacuole membrane [22]. Recent co-immunoprecipitation experiments of MYR1 also demonstrated TgPPM3C enrichment in RH strain parasites [21], in agreement with these TgPPM3C-HA Co-IP results. Despite the reproducible enrichment of a few proteins known to be secreted into the parasitophorous vacuole, we believed it was possible that the majority of TgPPM3C substrates were not identified by the Co-IP approach, particularly if de-phosphorylation events occur rapidly with transient interactions between TgPPM3C and phosphoprotein substrates.

## Phosphoproteomic comparison of ΔTgPPM3C and TgPPM3C-HA parasite cultures reveal putative phosphoprotein substrates

PP2C-class phosphatases identified in various organisms thus far exhibit no consensus target motif that allows for *a priori* PP2C substrate prediction. Therefore, we turned to a label-free phosphoproteomic approach to gain insights on putative TgPPM3C substrates in an unbiased fashion. HFF monolayers grown in 15cm diameter culture dishes were heavily infected in triplicate at a multiplicity of infection of five (MOI 5) with either TgPPM3C-HA (referred to as wild type, or "WT", for the purposes of this experiment,) or ΔTgPPM3C parasites and allowed to grow under tachyzoite growth conditions for 36 hours, allowing for large vacuoles to be obtained prior to parasite egress. Protein was harvested from infected cultures using 5% SDS lysis buffer and digested using S-Trap columns (Protifi). Phosphopeptides were subsequently enriched with titanium dioxide (TiO$_2$) beads and analyzed by LC-MS/MS to obtain peptide spectra. To normalize phosphopeptide abundance by protein abundance, the flow-through fraction from TiO$_2$ enrichment (i.e. non-phosphorylated proteins) were collected from each replicate and also analyzed by LC-MS/MS.

LC-MS/MS analysis yielded a robust, high confidence detection of 3,986 phosphopeptides from both host cell and parasite (S2 Data). Among the 1,753 parasite phosphopeptides identified, 128 phosphopeptides were found to have significantly altered abundance in the ΔTgPPM3C strain, as defined by a p < 0.05 cut-off and log$_2$ fold-change of less than -1.5 or greater than 1.5 (Fig 3A, red points). Of these, 118 total phosphopeptides were significantly more abundant in the ΔTgPPM3C strain compared to the parental strain, whereas only 10 phosphopeptides, largely from proteins of unknown function, were significantly less abundant (Fig 3A). Interestingly, more than half (73) of all differentially abundant phosphopeptides belong to proteins previously known to be secreted into the lumen of the parasitophorous

**Table 1. Five parasite secreted proteins are significantly enriched by TgPPM3C-HA Co-IP.**

| ToxoDB ID (TgME49_) | Short Name | Fold-Enrichment, p-value Experiment 1 | Fold-Enrichment, p-value Experiment 2 |
|---|---|---|---|
| 270320 | TgPPM3C | 7.9, <0.0001 | 5.3, <0.0001 |
| 270240 | MAG1 | 4.7, <0.0001 | 3.2, <0.0001 |
| 254470 | MYR1 | 3.7, <0.0001 | 3.3, <0.0001 |
| 251540 | GRA9 | 4.0, <0.0001 | 1.9, 0.0013 |
| 203310 | GRA7 | 1.6, 0.0104 | 2.7, 0.0002 |

Two independent Co-IP experiments were performed to identify TgPPM3C-HA interacting proteins. Fold enrichment compared to eluates from untagged control samples and p-values from each experiment are provided. Only protein hits with greater than 1.5 fold enrichment compared control and p-values < 0.05 in both experiments are shown

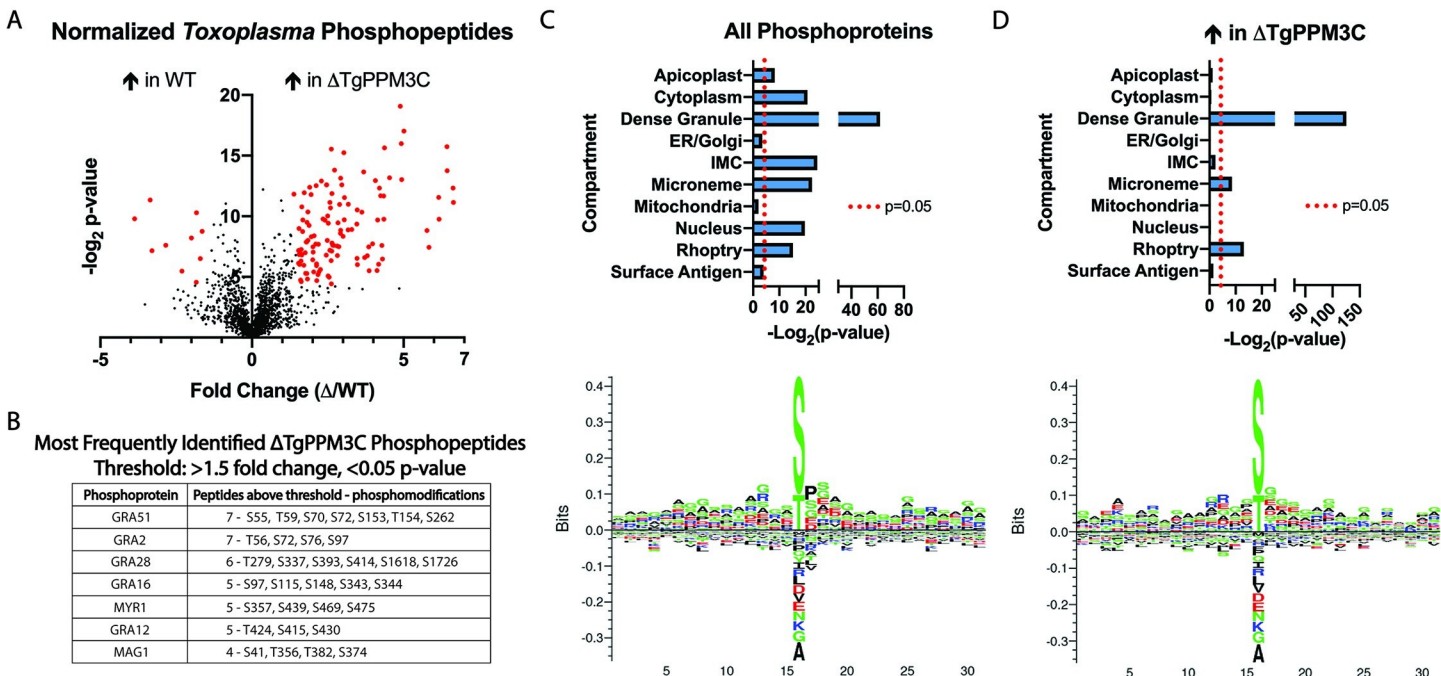

**Fig 3. Phosphoproteomic analysis of TgPPM3C-HA (WT) and ΔTgPPM3C infected cultures identifies phosphopeptides that are more abundant in ΔTgPPM3C cultures. (A)** Volcano plot depicting significant fold changes in *Toxoplasma* phosphopeptides from ΔTgPPM3C samples compared to TgPPM3C-HA (WT) samples. Phosphopeptides with fold changes greater than 1.5 or less than -1.5 and with -log$_2$ p-values greater than 4.32 (equivalent to $p < 0.05$) are highlighted in red. Using this criteria, 118 phosphopeptides are more abundant in ΔTgPPM3C cultures, as opposed to 10 phosphopeptides more abundant in WT cultures, suggesting that the absence of TgPPM3C predominately results in the accumulation of phosphoproteins that are normally dephosphorylated by TgPPM3C. Phosphopeptides are normalized to their respective protein abundance, detected in the flow-through fraction of titanium dioxide beads used for phosphopeptide enrichment. See Dataset S2 for specific phosphopeptide fold changes and p-values.**(B)** Phosphoproteins detected in this dataset with the most phosphopeptides above a 1.5 fold change and $p < 0.05$ threshold. Notably, two exported effectors (GRA16 and GRA28) and a protein involved in facilitating effector export (MYR1) are among these proteins. **(C)** Top–Results from hypergeometric testing to identify significant enrichment for subcellular parasite compartments among all phosphopeptides detected from wild-type and ΔTgPPM3C samples, compared to the entire *Toxoplasma* proteome. Various compartments are significantly enriched. Bottom–Amino acid motif, generated by Seq2Logo [57], derived from randomly selected phosphopeptides identified in wild-type and ΔTgPPM3C samples using a -15 and + 15 amino acid window with respect to a phosphoserine or phosphothreonine residue. As expected, no clear consensus motifs are evident. **(D)** As described in **(C)**, except that hypergeometric testing (top) and amino acid motif generation (bottom) was performed solely with phosphopeptides identified as significantly enriched in ΔTgPPM3C samples. A clear enrichment for the dense granule compartment and to a lesser extent rhoptry and microneme compartments are observed among these phosphopeptides, although no consensus amino acid motif is evident.

vacuole (S2 Data), suggesting TgPPM3C predominately affects the phosphorylation status of vacuolar proteins. To test this finding further, hypergeometric testing was performed to identify parasites compartments that were significantly enriched by all detected phosphopeptides from both wild-type and ΔTgPPM3C samples as compared to the entire *Toxoplasma* proteome (Fig 3C, upper panel). Enrichment could be seen for several parasite compartments, including compartments containing non-secreted proteins such as the nucleus and cytoplasm (Fig 3C, upper panel). In contrast, among the phosphopeptides significantly more abundant in ΔTgPPM3C samples, dense granule, rhoptry, and micronemes were the only significantly enriched compartments (Fig 3D, upper panel). The most robust compartment enrichment observed for this analysis was the dense granule, suggesting that the majority of phosphoproteins significantly affected by the absence of TgPPM3C are GRA proteins secreted into the parasitophorous vacuole (Fig 3D, upper panel). Among the phosphopeptides that were significantly more abundant in ΔTgPPM3C samples, no amino acid motif appeared to be enriched when inspected within a -15 and +15 amino acid window (Fig 3D, lower panel) as compared to -15 and +15 amino acid window analyses of randomly selected *Toxoplasma*

phosphopeptides (Fig 3C, lower panel). This result is in agreement with the notion that PP2C phosphatases do not have clear amino acid target motifs that are recognized and dephosphorylated, although it is conceivable that a consensus sequence was not readily apparent in this dataset due to the presence of phosphopeptides that are both significancy more abundant in ΔTgPPM3C samples and are not direct TgPPM3C substrates.

## GRA16 and GRA28 export from the parasitophorous vacuole is impaired in ΔTgPPM3C parasites

In line with our initial hypothesis that TgPPM3C influences the phosphorylation status of vacuolar proteins, we focused our studies on phosphoproteins identified by LC-MS/MS with phosphopeptides that were significantly more abundant in ΔTgPPM3C cultures and that were predicted or confirmed vacuolar proteins. Intriguingly, this list was well-represented by phosphopeptides belonging to two proteins known to be exported beyond the parasitophorous vacuole into the host cell nucleus, GRA16 [23] and GRA28 [24], as well as the afore-mentioned MYR1 protein (Fig 3B). Based on these observations, and the detection of MYR1 from TgPPM3C Co-IP experiments (Table 1), we next evaluated whether defects in effector export were evident in the ΔTgPPM3C strain. Transient transfections with plasmids encoding epitope tagged exported effectors GRA16, GRA24, GRA28, and TgIST under control of their endogenous promoters were performed with PruQ and ΔTgPPM3C parasites, and host nuclear accumulation for each effector was assessed 24 hours post-infection. A significant reduction in GRA16 and GRA28 nuclear accumulation was observed during infection with the ΔTgPPM3C strain as compared to the PruQ strain, likely indicating defects in effector export (Fig 4A). Interestingly, no significant defects in the host nuclear accumulation of GRA24 or TgIST were observed, indicating that a global reduction of effector export did not seem to be occurring in the vacuoles of ΔTgPPM3C parasites (Fig 4A).

HFF monolayers were next infected with either PruQ, ΔTgPPM3C, or TgPPM3C-COMP parasites and assayed for alterations in host c-Myc upregulation, which is known to be dependent on MYR1 and GRA16 export [22,25]. The results demonstrated a significant reduction in host c-Myc upregulation during infection with ΔTgPPM3C parasites compared to the PruQ and TgPPM3C-COMP strains (Fig 4B). Notably, all strains significantly induced host c-Myc upregulation compared to uninfected serum-starved fibroblasts, indicating that the absence of TgPPM3C does not result in a complete loss of GRA16 translocation and c-Myc induction, in agreement with recent observations made by Cygan et al. on ΔTgPPM3C mutants [21]. We also probed for alterations in host IRF1 upregulation, which is known to be inhibited in a TgIST-dependent fashion in infected cells stimulated with IFN-γ [26,27], as well as for alterations in host nuclear EZH2 induction, known to be dependent on the MYR1-dependant effector TEEGR/HCE1 [28,29]. The results showed no significant differences in host IRF1 attenuation following IFN-γ stimulation (Fig 4C) or in host cell EZH2 induction when comparing PruQ or TgPPM3C-HA, ΔTgPPM3C, and TgPPM3C-COMP infections (Fig 4D), providing further evidence that the export of only certain MYR1-dependant effector proteins (GRA16 and GRA28) appears to be impaired in the ΔTgPPM3C strain.

## Endogenous GRA16 export is perturbed in ΔTgPPM3C parasites

We next tested whether export defects could be observed after epitope-tagging the endogenous locus of GRA16 in the ΔTgPPM3C strain, comparing this strain to an endogenously tagged GRA16-3xHA PruQ strain [30]. A significant reduction in host nuclear GRA16 accumulation was observed after 24 hours of infection with the ΔTgPPM3C strain compared to the PruQ strain (S3A Fig), confirming that the apparent GRA16 export defect in ΔTgPPM3C parasites

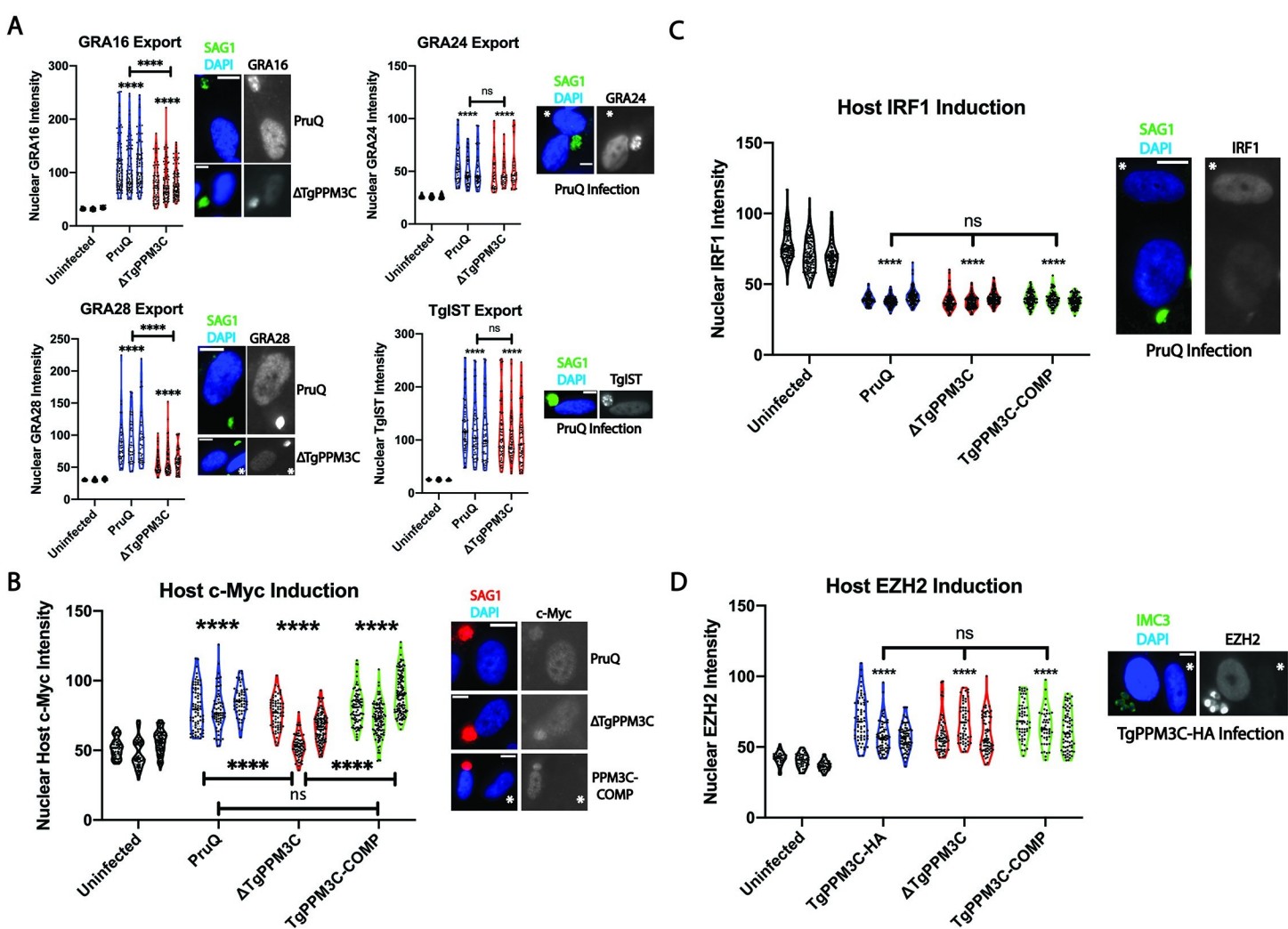

**Fig 4. ΔTgPPM3C parasites exhibit partial defects in protein effector export. (A)** Violin plots of GRA16-3xHA, GRA24-3xHA, GRA28-3xHA, and TgIST-3xHA accumulation in host nuclei infected with either PruQ or ΔTgPPM3C parasites, transiently transfected with epitope tagged effector constructs and allowed to infect HFF monolayers for 24 hours. Effector protein fluorescence was quantified from fibroblast nuclei containing a single HA-positive parasite vacuole. Violin plots of uninfected host nuclei in the same monolayer are provided to demonstrate background. A significant decrease in GRA16 and GRA28, but not GRA24 and TgIST, host nuclear accumulation is observed during ΔTgPPM3C infection, likely indicating defects in effector export from the parasitophorous vacuole. Violin plots represent three independent experiments. Representative images from which effector intensity was quantified are shown on the right. Antibody to SAG1 was used as a parasite marker, DAPI as a host nucleus marker, and anti-HA antibody was used to detect epitope tagged effectors. The white asterisks indicate uninfected fibroblast nuclei. **(B)** Violin plots of c-Myc induction in host nuclei infected with either PruQ, ΔTgPPM3C, or TgPPM3C-COMP parasites for 24 hours. Host c-Myc fluorescence was quantified from fibroblast nuclei containing a single parasite vacuole. A significant decrease in host c-Myc induction is observed during ΔTgPPM3C infection compared to PruQ and TgPPM3C-COMP infections. Notably, significant induction of host c-Myc expression over uninfected host cells are observed during infection with all three strains (asterisks above each plot). Violin plots represent three independent experiments. Representative images from which c-Myc intensity were quantified are shown on the right. Antibody to SAG1 was used as a parasite marker and DAPI as a host nucleus marker. Violin plots of uninfected host nuclei in the same monolayer are provided to demonstrate background. The white asterisk indicates an uninfected fibroblast nucleus. Non-specific labeling of parasite vacuoles can be observed with the c-Myc antibody used to label host cells. **(C)** Violin plots of IRF1 induction in host nuclei infected with either PruQ, ΔTgPPM3C, or TgPPM3C-COMP parasites and stimulated with IFN-γ for six hours prior to fixation at 24 hours post-infection. Host IRF1 fluorescence was quantified from fibroblast nuclei containing a single parasite vacuole. Although each strain significantly attenuated IRF1 induction compared to uninfected cells (asterisks above each plot), no significant differences in the extent of IRF1 suppression are observed between any of the strains. Violin plots of uninfected host nuclei in the same monolayer are provided to demonstrate background. Violin plots represent three independent experiments. A representative image from a PruQ infected fibroblast and uninfected fibroblast (white asterisk) is shown beside the violin plots. Antibody to SAG1 was used as a parasite marker and DAPI as a host nucleus marker. **(D)** Violin plots of EZH2 induction in host nuclei infected with either TgPPM3C-HA, ΔTgPPM3C, or TgPPM3C-COMP parasites for 24 hours. Host EZH2 fluorescence was quantified from fibroblast nuclei containing a single parasite vacuole. Each strain significantly induces host EZH2 compared to uninfected cells (asterisks), although no significant differences in host EZH2 induction were observed between any of the strains. Violin plots of uninfected host nuclei in the same monolayer are provided to demonstrate background. Violin plots represent three independent experiments. A representative image from a TgPPM3C-HA infected fibroblast and uninfected fibroblast (white asterisk) is shown beside the violin plots. Antibody to IMC3 was used as a parasite marker and DAPI as a host nucleus marker. The white asterisk indicates an uninfected fibroblast nucleus. Non-specific parasite labeling is evident with the EZH2 antibody used to label host cells. For all violin plots in this figure, **** asterisks indicate $p < 0.0001$ and ns indicates no significant difference. Kruskal-Wallis and Dunn's multiple comparisons tests were performed to calculate p-values. All scale bars in this figure indicate 10μm.

holds true for both endogenous protein and GRA16 transiently ectopically expressed (Fig 4A). To determine whether a phosphorylated GRA16 species might be more abundant in the ΔTgPPM3C background, we also probed for GRA16 migration by immunoblotting protein lysates from the endogenously tagged GRA16-3xHA PruQ and ΔTgPPM3C strains. Protein was harvested in the presence of phosphatase inhibitors from tachyzoite infected monolayers 24 hours post-infection. Two GRA16-3xHA bands were readily apparent by immunoblotting in both strains; one band migrating between 55kDa and 40kDa, and a larger band migrating between 70kDa and 55kDa (S3B Fig). The predicted molecular weight of mature ASP5-cleaved GRA16-3xHA protein [31] is ~51kDa, which might be represented by the smaller of the two bands in both PruQ and ΔTgPPM3C strains. The larger band was affected by alkaline phosphatase (AP) treatment in both PruQ and ΔTgPPM3C backgrounds (S3B Fig, +AP), indicating that these protein bands likely represent phospho-GRA16, although the phosphatase treatment may have been insufficient in fully de-phosphorylating GRA16 in both samples. Interestingly, the >55kDa band in the ΔTgPPM3C strain appears to migrate slightly slower than the >55kDa band in the PruQ strain, suggesting that GRA16 is hyperphosphorylated in the ΔTgPPM3C strain in agreement with the phosphoproteomic dataset. Both GRA16 species detected in infected monolayer lysates are also readily detectable in extracellular tachyzoite lysates, suggesting that GRA16 may be phosphorylated within the parasite prior to secretion into the parasitophorous vacuole (S3B Fig, extracellular parasites panel).

## Phosphomimetic mutations of GRA16 recapitulate vacuolar export defects

The export defects observed in the ΔTgPPM3C strain suggested that GRA16 and GRA28 export are likely impaired due to their altered phosphorylation status, although the altered phosphorylation status of specific MYR1 residues could also be contributing to the observed export defects. To test whether only the phosphorylation status of GRA16 influences export from the parasitophorous vacuole, we mutagenized two serine residues in the GRA16 protein (S97 and S148) whose phosphorylation was robustly detected in ΔTgPPM3C parasites and absent in TgPPM3C-HA parasites (Fig 3B, S2 Data). These residues fall within the R1 and R2 tandem repeat regions (R1: 59-100aa, R2: 125-168aa) previously shown to be critical for GRA16 export [23,31]. Phosphomimetic (S97E/S148E) and phosphoablative (S97P/S148P) mutations were introduced into constructs encoding epitope-tagged GRA16 under control of the endogenous GRA16 promoter (Fig 5A). For phosphoablative mutations, proline substitutions were used instead of alanine so as to maintain protein intrinsic disorder within the mutant GRA16 sequence, which is known to be a critical property that allows for GRA16 export [31]. Proline, unlike alanine, promotes protein disorder and is commonly found within intrinsically disordered proteins [32]. Following transfection of PruQ parasites, host nuclear GRA16 accumulation was assessed by immunofluorescence 24 hours post-infection with polyclonal parasites expressing either wild type (unmodified), phosphomimetic, or phosphoablative GRA16 constructs. The results demonstrated that both mutant GRA16 proteins were significantly less abundant within infected host cell nuclei compared to the wild type GRA16 protein, as measured by immunofluorescence intensity (Fig 5B). Phosphomimetic GRA16 was found to be significantly less abundant in host nuclei compared to phosphoablative GRA16 (Fig 5B), indicating that the glutamate substitutions compromised GRA16 export more drastically than the proline substitutions.

The findings on mutant GRA16 in the PruQ strain suggest that the reduced nuclear accumulation of GRA16, that was most evident for phosphomimetic GRA16, was most likely due to a decrease in effector export; however, an alternative hypothesis is that this could be due to mutant GRA16 protein being inherently less stable either in the host nucleus or

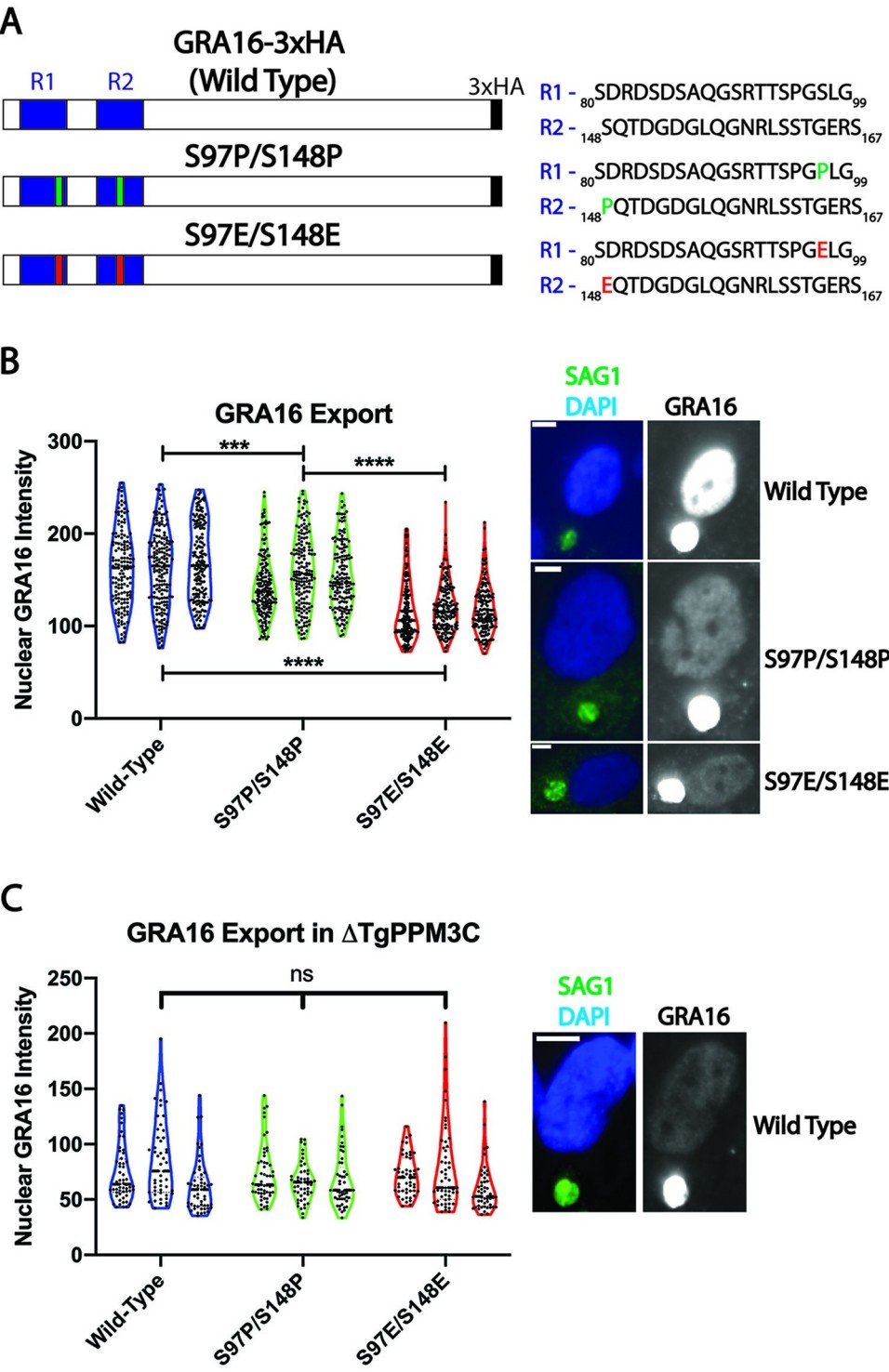

**Fig 5. Phosphomimetic mutations impair the export of GRA16 from the parasitophorous vacuole. (A)** Schematic of GRA16-3xHA constructs, including the partial sequence and location of previously described tandem repeat regions (R1 and R2). Mutations were designed to introduce either proline (S97P/S148P) or glutamate (S97E/S148E) in place of serine 97 and serine 148. **(B)** Violin plots and representative images of GRA16 intensity in host nuclei infected with parasites expressing either wild type, S97P/S148P, or S97E/S148E GRA16-3xHA constructs. Nuclear GRA16 intensity was quantified from fibroblast nuclei containing a single HA-positive parasite vacuole. A significant decrease in nuclear S97P/S148P and S97E/S148E GRA16 intensity is observed compared to wild type GRA16 intensity, while S97E/S148E GRA16 nuclear intensity values were also found to be significantly less intense compared to values

recorded from S97P/S148P infections. Violin plots represent three independent experiments. Representative images from which GRA16 intensities were quantified are shown to the right. Antibody to SAG1 was used as a parasite marker and DAPI as a host nucleus marker. *** asterisks indicate p < 0.001, **** indicates p < 0.0001. Kruskal-Wallis and Dunn's multiple comparisons tests were performed to calculate p-values. Scale bars indicate 5μm in **B** and 10μm in **C**. (**C**) Violin plots and representative images of GRA16 intensity in host nuclei as described in (**B**), except that all experiments were conducted in the ΔTgPPM3C strain. No significant differences in host nuclear intensity are observed between any of the GRA16 constructs during ΔTgPPM3C infection, suggesting that wild type and mutagenized GRA16 have seemingly similar stabilities during parasite infection.

parasitophorous vacuole compartments compared to wild type GRA16 protein. To account for this possibility, we transfected the same mutagenized and wild-type GRA16 constructs into ΔTgPPM3C parasites, reasoning that if the defect were not occurring specifically during the export process across the parasitophorous vacuole membrane, differences in GRA16 nuclear accumulation may still be evident in the absence of TgPPM3C. Quantifying host nuclear GRA16 accumulation after 24 hours of infection revealed no significant differences between unmodified and mutagenized GRA16 in the ΔTgPPM3C strain (Fig 5C), indicating that the stability of both mutagenized GRA16 variants, at least as measured by host nucleus accumulation, is similar to wild-type GRA16. Thus, we believe these data provide support for a model whereby TgPPM3C mediated dephosphorylation facilitates efficient GRA16 export into the host cell.

## Discussion

PP2C-class serine/threonine protein phosphatases are widespread throughout all domains of life. Through the regulation of protein phosphorylation status, PP2C phosphatases influence cellular activity in diverse fashions, ranging from cell growth to survival [33]. An expansion of genes encoding PP2C catalytic domains is noteworthy in plants, some of which play major roles in stress-response signaling pathways [34]. Intriguingly, *Toxoplasma* exhibits several plant-like features, such as the expression of calcium dependent protein kinases [35], plant-like AP2 transcription factors [36], as well as the expansion of genes predicted to encode PP2C-class phosphatases [15]. In good agreement with this observation is a previous study on phosphatase activity in *T. gondii* parasite lysates, which demonstrated that PP2C activity was more prevalent than PP2A activity as determined by the use of phospho-casein as a substrate and PP2A-class phosphatase inhibitors [37]. TgPPM3C is one of several PP2C genes predicted to encode a signal peptide, suggesting several PP2C phosphatases may enter the secretory pathway and be packaged into secretory organelles. Indeed, a previous study demonstrated that one of these PP2C phosphatases, dubbed PP2C-hn, is trafficked to the rhoptry organelle and secreted into the host cell nucleus at the time of invasion [38]. In contrast, we observed that TgPPM3C is seemingly packaged partially into dense granules (Fig 1B), the contents of which are continuously released into the parasitophorous vacuole during intracellular infection [39]. Transcriptomic data deposited on ToxoDB [40] demonstrate TgPPM3C mRNA transcripts in tachyzoites, bradyzoites, and merozoites, suggesting a shared role for this phosphatase across different *Toxoplasma* life stages. Both the catalytic domain and N-terminal extension of TgPPM3C appears to be largely conserved across three strains representative of three distinct lineages of *Toxoplasma* (GT1, ME49, and VEG; S4 Fig). Furthermore, a high degree of TgPPM3C homology is evident in *Neospora caninum* and *Hammondia hommondi* (S4 Fig), possibly indicating a common need for this particular PP2C phosphatase among the close relatives of *Toxoplasma*.

The conserved PP2C catalytic domain forms a tertiary structure in which a beta-sandwich surrounded by alpha helices forms a pocket that allows metal-coordinating and phosphate

binding residues to catalyze de-phosphorylation [41] (Fig 1A). In agreement with homology to the PP2C domain, the X-ray crystal structure of the TgPPM3C catalytic domain obtained by Almo and colleagues [42] demonstrates that the catalytic domain of TgPPM3C appears to adopt the appropriate tertiary structure to mediate de-phosphorylation (S5A and S5B Fig). A largely similar catalytic pocket architecture can be observed in TgPPM3C (S5A and S5B Fig) compared to human PPM1A (S5C and S5D Fig). Notably, although the TgPPM3C active site must still be considered putative, this region forms a robust acidic pocket that likely attracts $Mg^{2+}/Mn^{2+}$ cations (S5B Fig, bottom panel), as observed in human PPM1A (S5D Fig, bottom panel). The flap domain (green regions in S5 Fig) is a common feature among PP2C phosphatases that is thought to be involved in substrate preference by docking phosphopeptides along the groove formed between the active site and flap domain [41]. Indeed, this configuration has been demonstrated for the *Arabidopsis* phosphatase TAP38/PPH1, which demonstrates a preference for phospho-LHCII by virtue of basic residues adjacent to the phosphothreonine substrate, which mediate electrostatically favorable interactions with the acidic TAP38/PPH1 catalytic pocket [43]. TgPPM3C contains an additional cleft not observed in human PPM1A beside the flap domain and putative active site (S5A and S5B Fig, white region); this groove may also be involved in phosphoprotein substrate docking, although more experiments are needed to test this hypothesis.

Current data on PP2C substrate recognition suggest that no specific consensus motifs are recognized by the catalytic domain; rather, it is thought that interacting protein partners and/ or divergent targeting domains present in PP2C phosphatases direct the enzyme to substrates [44]. Stable interactions with proteins, such as the interaction detected between MYR1 and TgPPM3C in Cygan et al. [21], and/or the N-terminal extension of TgPPM3C may serve as entities responsible for targeting this phosphatase to the appropriate substrates. Along these lines, we observed the lack of a consensus motif among the phosphopeptides that were more abundant in ΔTgPPM3C cultures (Fig 3D, lower panel), although it is currently unclear what proportion of these phosphopeptides are direct TgPPM3C phosphoprotein substrates. More than half of these phosphopeptides belong to proteins known to be secreted into the parasitophorous vacuole, either in soluble or membrane bound forms (Fig 3A and S2 Data). Furthermore, the dense granule organelle was the most represented compartment when considering phosphopeptides that were significantly more abundant in ΔTgPPM3C cultures (Fig 3D, upper panel), although we recognize that a bias toward the enrichment of dense granule proteins is evident in our phosphopeptide dataset based on hypergeometric testing of all detected phosphopeptides (Fig 3C, upper panel). Interestingly, among the several GRA7 phosphopeptides identified, one phosphopeptide was significantly less abundant in ΔTgPPM3C cultures (T90, S2 Data), indicating TgPPM3C may affect the phosphorylation status of proteins indirectly through the regulation of kinases and/or other phosphatases. Indeed, phosphoproteomic analysis also identified a significant increase in one phosphopeptide from another putative PP2C class phosphatase, TgPPM3A (S167, S2 Data), predicted by LOPIT data on ToxoDB to localize to dense granules [40], indicating potential TgPPM3A secretion into the vacuole and cross-talk between PP2C phosphatases.

Phosphopeptides from GRA2 were frequently identified as more abundant in ΔTgPPM3C cultures, including peptides with phosphosites identified as down-regulated in ΔWNG1 cultures from a study on the vacuolar WNG1 kinase [13] (T56 and S72, Fig 3B). Although we did not find evidence for dysregulated GRA2 activity by inspecting the IVN of vacuoles formed by the ΔTgPPM3C strain (S2 Fig), we cannot exclude the possibility that abnormalities in ΔTgPPM3C parasitophorous vacuole architecture may be occurring at a different period during tachyzoite development or may be more evident in tissue cysts. We note that phosphopeptides from two recently characterized vacuolar strand-forming proteins, GRA29 and SFP1

[45], were detected as significantly more abundant in ΔTgPPM3C cultures, suggesting that these proteins may be substrates of TgPPM3C that are regulated by dephosphorylation events within the vacuolar compartment. Although Young et. al. observed no differential phosphorylation of GRA29 and SFP1 when comparing tachyzoite and *in vitro* bradyzoite cultures, their data demonstrate profound global differences in vacuolar phosphoprotein status when comparing these two life stages [45]. It will be interesting to explore the possibility that TgPPM3C activity might be regulated in a stage specific manner, perhaps via protein interactions with different targeting factors present in developing tissue cysts but absent in tachyzoite vacuoles.

The export of protein effectors from the *Toxoplasma* parasitophorous vacuole is a fast-developing area of inquiry [10]. Exported effectors such as GRA16 and TgIST have been elegantly characterized and shown to influence host cell signaling in profound ways that benefit the intracellular development of the parasite [23,26,27]. Several effectors exported from the parasitophorous vacuole have been described apart from those explored in this study [9,28,29,46], while the list of proteins necessary for the export of these proteins has also been growing, including an acid phosphatase domain-containing protein in GRA44 [21,47–49]. Cygan et. al. recently reported no significant growth defect in RH strain ΔTgPPM3C parasites and an interaction between MYR1 and TgPPM3C as ascertained by Co-IP. Using a qualitative assay for the presence or absence of host c-Myc induction and GRA16 export there was no observed difference in the wild type RH strain compared to ΔTgPPM3C RH strain parasites [21]. The growth defect we observed in Pru strain ΔTgPPM3C parasites suggests that this type II strain relies more heavily on the processes regulated by TgPPM3C compared to the more virulent type I RH strain. While our findings are in agreement with Cygan et. al. [21] that TgPPM3C is not essential for effector export, the quantitative data indicate that TgPPM3C influences the amount or efficiency of export of GRA16 and GRA28 effectors, and by extension the induction of host c-Myc during intracellular infection (Fig 4A and 4B). Importantly, the absence of TgPPM3C does not appear to affect vacuolar protein export globally (Fig 4), indicating that TgPPM3C likely facilitates the export of only a specific set of effector proteins. Similarly, we speculate that the MYR1 phospho-modifications detected as more abundant in the ΔTgPPM3C strain may be involved in contacts with only select effector proteins (e.g. GRA16 and GRA28) and/or potentially non-effector proteins perhaps in the assembly of the translocon complex.

Based on the impaired export of phosphomimetic GRA16, we propose that the de-phosphorylation of GRA16 somehow enables this effector to traverse the putative vacuolar protein translocon more efficiently. However, if de-phosphorylation is necessary for efficient export of some effector proteins, it is unclear what purpose phosphorylation serves prior to export. All effector proteins exported from the parasitophorous vacuole thus far are predicted to exhibit a large degree of protein intrinsic disorder; a property that is highly correlated with the propensity of proteins to aggregate and form a phase separated state [50]. Perhaps by introducing negatively charged regions that repel each other, phosphorylation of intrinsically disordered effector proteins prevent their aggregation. Precedence for this notion can be found in the P granules of *C. elegans* embryos, in which phosphorylation of intrinsically disordered proteins drives de-formation of phase separated granules whereas phosphatase activity induces the opposite effect [51], although a key feature of P granule formation is the involvement of RNA as a seed for intrinsically disordered proteins with RNA binding motifs to form a phase separated state. The protein GRA45 was recently shown to facilitate effector protein export by seemingly preventing the aggregation of vacuolar GRA16 and GRA24 [49], providing support for the notion that effector proteins can aggregate within the parasitophorous vacuole. Intriguingly, in RS-repeat rich proteins, the phosphorylation of multiple serine residues has been shown to induce a more rigid and less disordered protein structure [52]. Given that the export of S97P/S148P GRA16 was also slightly impaired, and that proline substitutions should in

principle maintain protein intrinsic disorder, we speculate that certain non-phosphorylated serine motifs are preferentially recognized by the vacuolar translocon, as opposed to a model where the de-phosphorylation of serine residues serve as a switch from an ordered-to-disordered protein state. Experiments probing the tertiary structure (or lack thereof) of GRA16 are needed to fully understand the structural consequences of phosphorylation in this effector protein, particularly within the R1 and R2 tandem repeat regions.

Altogether, we propose a model whereby TgPPM3C, following secretion into the parasitophorous vacuole, de-phosphorylates several vacuolar phosphoprotein substrates such as GRA16 and GRA28, allowing these and possibly other effectors to be efficiently exported and proceed with host cell manipulation, thus optimizing parasite growth and virulence. To further validate this model, more investigations are needed on TgPPM3C catalytic activity and substrate preference, where TgPPM3C activity is detected *in situ*, and what other consequences arise from the lack of de-phosphorylation in vacuolar proteins not explored in this study.

## Materials and methods

### Ethics statement

All mouse experiments were conducted according to guidelines from the United States Public Health Service Policy on Humane Care and Use of Laboratory Animals. Animals were maintained in an AAALAC-approved facility, and all protocols were approved by the Institutional Care Committee of the Albert Einstein College of Medicine, Bronx, NY (Animal Protocol 20180602; Animal Welfare Assurance no. A3312-01).

### Cell culture

PruΔku80Δhxgprt LDH2-sfGFP parasites [53] (i.e. PruQ parasites) were continuously passaged in human foreskin fibroblasts (HFF:ATCC:CRL-1634; Hs27) in a 37˚C, 5% $CO_2$ incubator using Dulbecco's Modified Eagle Media (DMEM, Gibco) supplemented with 10% fetal calf serum, 1% L-glutamine, and 1% penicillin and streptomycin. Cultures were regularly inspected and tested negative for mycoplasma contamination. Bradyzoite induction was performed at the time of invasion by replacing growth media with bradyzoite induction media (50 mM HEPES, pH 8.2, DMEM supplemented with 1% FBS, penicillin and streptomycin) prior to infection of HFFs with egressed tachyzoites. Bradyzoite induced cultures were maintained in a 37˚C incubator without $CO_2$, with induction media replaced every 2 days.

### Cloning and parasite transfections

For a full list of oligonucleotides used for cloning and genetic manipulations, refer to S1 Table. Briefly, for TgPPM3C epitope tagging, a single guide RNAs (sgRNA) targeting the C-terminus of the TgPPM3C gene was cloned into the p-HXGPRT-Cas9-GFP plasmid backbone using KLD reactions (New England Biolabs), as previously described [54]. Donor sequences for homology mediated recombination were generated by amplifying a 1xHA tag, the 3'UTR of HXGPRT, and a DHFR mini-cassette to confer pyrimethamine resistance from the previously described pLIC-3xHA-DHFR plasmid backbone [20]. Primers used to amplify this donor sequence contained overhangs with 40bp homology to the C-terminus and 3'UTR, as well as mutations in the Cas9 target site to prevent re-targeting by Cas9. To knockout the TgPPM3C gene (and generate ΔTgPPM3C strains), a sgRNA targeting the N-terminus of TgPPM3C was designed and cloned into the afore-mentioned Cas9 plasmid backbone. Donor sequences containing a multi-stop codon sequence and lacking the TgPPM3C start codon were designed for co-transfection with the TgPPM3C knockout sgRNA. To reintroduce expression of TgPPM3C in the knockout strain (and generate TgPPM3C-COMP

strains), sgRNA targeting the multi-stop codon sequence was designed and introduced into a Cas9 plasmid as described above. Donor sequences were designed to restore the original coding sequence of TgPPM3C when co-transfected with the sgRNA used for complementation.

The GRA16, GRA24, GRA28, and TgIST loci were amplified from PruΔku80Δhxgprt genomic DNA with primers to the coding sequence of each gene and 1.5kb upstream of the start codon, with overhangs to pLIC-3xHA-DHFR sequences. Gibson Assemblies (NEBuilder HiFi DNA Assembly) were subsequently performed to clone each gene into PCR amplified pLIC-3xHA-DHFR plasmid backbones. To mutagenize GRA16 serine residues, KLD reactions were performed using the pLIC-GRA16-3xHA plasmid as template DNA for PCR, with primers designed to introduce mutations as intended.

For each transfection, $5x10^6$ to $1x10^7$ PruΔku80Δhxgprt tachyzoites were electroporated in cytomix [55] after harvesting egressed parasites from HFF monolayers and filtering through 5μm filters. Selection of transfected parasites was performed either with 2μM pyrimethamine for at least two passages or with media containing 25μg/mL mycophenolic acid and 50μg/mL xanthine 24 hours post-transfection for 6 days before removing selection media and subcloning by limiting dilution after sufficient parasite egress was observed. For Cas9 transfections, 7.5μg of uncut Cas9 plasmid and 1.5μg of PCR amplified donor sequence or 280 pmol unannealed donor sequences were used for each transfection. Cas9 transfections targeting the C-terminus of GRA16 were performed using constructs described in [30]. For transient transfections of PruΔku80Δhxgprt parasites with plasmids encoding GRA16-3xHA, GRA24-3xHA, GRA28-3xHA, and TgIST-3xHA, 50μg of uncut plasmid was used. For transfections with mutagenized and wild type GRA16-3xHA plasmids, 10μg of uncut plasmid were used followed by 2μM pyrimethamine selection as described above.

## Immunofluorescence assays

HFF monolayers were grown to confluency on glass coverslips and infected with egressed tachyzoites at an MOI of 1 for most immunofluorescence assays. For IFN-γ stimulation experiments, infected HFF monolayers were stimulated with 100U/mL recombinant human IFN-γ (R&D Systems) 24 hours post-infection and fixed 6 hours post-stimulation. All coverslips were fixed with 4% PFA for 20 minutes at room temperature, permeabilized in a 0.2% Triton X-100, 0.1% glycine solution for 20 minutes at room temperature, rinsed with PBS, and blocked in 1% BSA for either 1 hour at room temperature or at 4°C overnight. Coverslips were labeled with antibodies as follows: HA-tagged proteins were detected with rat anti-HA 3F10 (Sigma 1:200), parasite cyst wall and parasite cytoplasm by in-house mouse SalmonE anti-CST1 (1:500) and rabbit anti-TgALD1 (1,500, kind gift from Dr. Kentaro Kato) respectively, host c-Myc with rabbit anti-c-Myc (D84C12, Cell Signaling, 1:250), IRF1 by rabbit anti-IRF1 (D5E4, Cell Signaling, 1:500), tachyzoite SAG1 by mouse anti-SAG1 (Thermo Fisher, 1:500), parasite TgIMC3 by rat anti-TgIMC3 (1,2000, kind gift from Dr. Marc Jan-Gubbels), and GRA6 by anti-HF10 (1,1000, kind gift from Dr. Nicolas Blanchard). Secondary antibodies conjugated to Alexa Fluorophores 488, 555, 594, and 633 targeting each primary antibody species were used at a dilution of 1:1000 (Thermo Fisher). DAPI counterstain was used to label parasite and host cell nuclei (1,2000). Coverslips were mounted in ProLong Gold Anti-Fade Reagent (Thermo) and imaged using either a Leica SP8 confocal microscope, a Nikon Eclipse widefield fluorescent microscope (Diaphot-300), or a Pannoramic 250 Flash III Automated Slide Scanner (3D Histech).

## Quantitative image analysis

For all fluorescence quantification, at least 30 randomly selected fields of view from every labeled coverslip was exported from CaseViewer (3D Histech) software after acquiring images

with a Pannoramic 250 Automated Slide Scanner using a 40X objective lens. Identical exposure times were used for each independent experiment to detect and subsequently compare fluorescence intensity values. Exported images were blinded prior to ImageJ analysis (NIH), segmenting regions of interest (ROI) manually from fibroblasts that contained individual parasitophorous vacuoles. The mean gray value was measured from each ROI in the fluorescent channel used to detect the signal of interest. Mean gray values were plotted using PRISM 8 (GraphPad). As each mean gray value dataset did not exhibit a normal distribution, nonparametric Kruskal-Wallis tests and Dunn's multiple comparisons test in PRISM software were used to compare the means from three independent experiments between groups and determine statistical significance.

### Invasion assay

Parasites were allowed to infect HFF monolayers grown on glass coverslips in 24-well plates for 30min in 37C and 5% $CO_2$, using an MOI of 5. After removing the supernatant, monolayers were washed five times with PBS and fixed in 4% PFA. Fixed coverslips were blocked with 1% BSA/PBS before labeling extracellular parasites with mouse anti-SAG1. Post-SAG1 labeling, monolayers were fully permeabilized with 0.2% Triton and blocked again in 1% BSA before labeling extracellular and intracellular parasites with in-house rabbit anti-Toxoplasma polyclonal antibody. Anti-mouse 488 and anti-rabbit 594 secondary antibodies were used to distinguish dual labeled extracellular parasites from intracellular parasites only labeled with rabbit anti-Toxoplasma. At least 100 parasites from 10 different fields of view were counted for each parasite strain per experiment.

### Replication assay

HFF monolayers grown on glass coverslips in 24-well plates were infected with parasites at an MOI of 1 in a 37C 5% $CO_2$ incubator. 32 hours post-infection, coverslips were fixed with 4% PFA, permeabilized with 0.2% Triton X-100, and labeled with anti-SAG1 to identify the number of parasites per vacuole by immunofluorescence. At least 150 vacuoles from 15 different fields of view containing either 2, 4, 8, or 16 parasites were assessed for each parasite strain per experiment.

### Egress assay

HFF monolayers grown on glass coverslips in 24-well plates were infected with parasites at an MOI of 1 in a 37C 5% $CO_2$ incubator. 36 hours post-infection, media was aspirated and coverslips rinsed twice with PBS. Coverslips were then treated with egress assay buffer (Hank's Balanced Salt Solution supplemented with 1mM $CaCl_2$, 1mM $MgCl_2$, and 10mM HEPES) containing either 2μM A23187 calcium ionophore or an equivalent volume of DMSO (control) and incubated for 3 minutes in the 37C 5% $CO_2$ incubator. Egress assay buffer was removed and coverslips were fixed with 4% PFA, permeabilized with 0.2% Triton X-100, and antibody labeled with mouse anti-SAG1 and rabbit anti-GRA6 (anti-HF10) along with appropriate secondary antibodies. At least 200 vacuoles from 20 different fields of view for each parasite strain per experiment were counted as "unoccupied" (GRA6 positive, SAG1 negative) or "occupied" (GRA6 positive and SAG1 positive). Only vacuoles containing at least four parasites were scored as "occupied".

### Plaque assay

Parasites were harvested from host cells with a 27G needle and filtered through a 5μm filter to remove host cell debris. Parasite numbers were counted with a hemocytometer, and 100

parasites from each strain were added in triplicate to wells containing confluent HFFs in 6-well dishes. Parasites were grown for 14 days before fixing and staining with a 20% methanol-0.5% crystal violet solution. Images collected from stained monolayers were blinded prior to analysis using ImageJ, separating merged neighboring plaques with the line tool prior to calculating plaque size. Kruskal-Wallis tests with Dunn's multiple comparisons were performed using the means from three independent experiments to determine significance with PRISM 8.

### Electron microscopy

For transmission electron microscopy, samples were prepared from human fibroblast monolayers infected with parasites grown under tachyzoite growth conditions for 32 hours. Cultures were fixed with 2.5% glutaraldehyde, 2% paraformaldehyde in 0.1 M sodium cacodylate buffer, post-fixed with 1% osmium tetroxide followed by 2% uranyl acetate, dehydrated through a graded series of ethanol and embedded in LX112 resin (LADD Research Industries, Burlington VT). Ultrathin sections were cut on a Leica Ultracut UC7, stained with uranyl acetate followed by lead citrate and viewed on a JEOL 1400EX transmission electron microscope at 80kv.

### Co-immunoprecipitation

In two separate experiments, 15cm diameter cell culture dishes containing confluent HFF monolayers were infected at an MOI of 3 with either TgPPM3C-HA or control PruQ non-HA tagged parasites. Dishes were washed with ice cold PBS 24 hours post-infection and lifted off each dish with a cell scraper in 1mL ice cold lysis buffer (50mM Tris pH 7.4, 200mM NaCl, 1% Triton X-100, and 0.5% CHAPS) supplemented with cOmplete EDTA-free protease inhibitor, (Sigma) and phosphatase inhibitors (5mM NaF, 2mM activated $Na_3VO_4$). Scraped cultures were passed through a 27G needle five times and sonicated for 30 seconds total (20% amplitude, 1 second pulses). Sonicated samples were incubated on ice for 30min, supernatant cleared by centrifugation (1000xg, 10min), and incubated overnight in a 4°C rotator with 1mg anti-HA magnetic beads (100uL slurry, Thermo Fisher). Following overnight incubation, beads were separated on a magnetic stand and washed twice in lysis buffer and four times in wash buffer (50mM Tris pH 7.4, 300mM NaCl, 0.1% Triton X-100) prior to elution in Laemmli buffer with 50mM DTT, boiling beads for 5min prior to magnetic separation and collection of eluate. Eluates were loaded, washed, and digested into peptides with 1μg of trypsin on S-TRAP micro columns (Protifi) per manufacturer guidelines. S-TRAP peptide eluates were concentrated with a speed vac, desalted in HLB resin (Waters), and concentrated in a speed vac once more prior to running through LC-MS/MS.

### Alkaline phosphatase treatment

HFF monolayers grown in T25 flasks were infected with either GRA16-3xHA tagged parasites or untagged control parasites for 24 hours at an MOI of 2 under tachyzoite growth conditions. Adherent cultures were rinsed twice in cold PBS and then harvested in 500μL of cold RIPA buffer supplemented with cOmplete EDTA-free protease inhibitor (Sigma). Protein lysates were sonicated (30 seconds total, 20% amplitude, 1 second pulses) and incubated on ice for 30 minutes to allow for solubilization. Aliquots of sonicated and solubilized lysates containing approximately 2μg total protein were either treated with 100U calf intestinal alkaline phosphatase (Promega) in alkaline phosphatase buffer (50mM Tris-HCl pH 9.3, 1mM $MgCl_2$, 0.1mM $ZnCl_2$, 1mM spermidine) or mock treated with alkaline phosphatase buffer only and incubated in a 37C water bath for 1 hour. The reaction was then terminated by the addition of Laemmli buffer, and samples were analyzed by SDS-PAGE and immunoblotting as described below.

## SDS-PAGE and immunoblotting

Protein lysates were prepared in radioimmunoprecipitation assay (RIPA) buffer after harvesting protein from infected fibroblasts cultures in T25 flasks 24 hours post-infection, or from 5μm filter purified extracellular parasites. Laemmli sample buffer was added to samples and boiled for 5–10 minutes before loading on SDS-PAGE 4–20% pre-cast gradient gels (TGX) and running gels for 1.5-2hrs at 100V. Transfer to PVDF membranes (Millipore) was performed in Towbin buffer (20% methanol, Tris/Glycine buffer) for 2 hours at 100V, and blocking in 5% BSA/TBST was performed overnight in 4˚C. Membranes were labeled in 5% BSA/TBST with anti-HA peroxidase conjugated antibody (Sigma, 1:200–1:1000), mouse anti-SAG1 (Thermo Fisher, 1:100), rabbit TgALD1 antibody (1:200), and anti-rabbit HRP antibodies (Thermo Fisher, 1:10000) followed by development of signal with West Pico Plus Chemiluminescent substrate, or by LiCor anti-rabbit 680 and LiCor anti-mouse 800 secondary antibodies. Images of labeled blots were collected with a Li-COR instrument (Odyssey Imaging System).

## Phosphoproteome preparation

15cm diameter cell culture dishes containing confluent HFF monolayers were infected in triplicate at an MOI of 5 with either TgPPM3C-HA or ΔTgPPM3C parasites. Dishes were washed with ice cold PBS 36 hours post-infection and lifted off each dish with a cell scraper in ice cold PBS supplemented with phosphatase inhibitors (5mM NaF, 2mM activated $Na_3VO_4$). Scraped cultures were pelleted at 3000rpm for 10min at 4˚C. Pellets were resuspended in 500μL S-Trap lysis buffer (5% SDS, 50mM TEAB, pH 7.5) supplemented with cOmplete EDTA-free protease inhibitor (Sigma) and HALT phosphatase inhibitor cocktail (Thermo Fisher). Lysates were sonicated (30 seconds, 20% amplitude, 1 second pulses) and cleared by centrifugation (13,000xg, 4˚C). Protein from the supernatant was quantified with a Micro BCA assay kit (Thermo Fisher) and 300μg of protein was loaded, washed, and digested with 3μg of trypsin on S-Trap mini columns (Protifi) per manufacturer guidelines. Phosphopeptides were enriched from 100μg of S-Trap peptide eluates using titanium dioxide beads ($TiO_2$, GL Sciences), as previously described [56]. Flow through from titanium dioxide bead enrichment was also collected for LC-MS/MS acquisition and normalization of phosphopeptide abundance by non-phosphopeptides. Following $TiO_2$ enrichment, peptides were concentrated with a speed vac, desalted in HLB resin (Waters), and concentrated in a speed vac once more prior to analyzing peptides by LC-MS/MS.

## LC-MS/MS acquisition and analysis

Samples were resuspended in 10 μl of water + 0.1% TFA and loaded onto a Dionex RSLC Ultimate 300 (Thermo Scientific, San Jose, CA, USA), coupled online with an Orbitrap Fusion Lumos (Thermo Scientific). Chromatographic separation was performed with a two-column system, consisting of a C18 trap cartridge (300 μm ID, 5 mm length) and a picofrit analytical column (75 μm ID, 25 cm length) packed in-house with reversed-phase Repro-Sil Pur C18-AQ 3 μm resin. Peptides were separated using a 120 min gradient from 2–28% buffer-B (buffer-A: 0.1% formic acid, buffer-B: 80% acetonitrile + 0.1% formic acid) at a flow rate of 300 nl/min. The mass spectrometer was set to acquire spectra in a data-dependent acquisition (DDA) mode. Briefly, the full MS scan was set to 300–1200 *m/z* in the orbitrap with a resolution of 120,000 (at 200 *m/z*) and an AGC target of 5x10e5. MS/MS was performed in the ion trap using the top speed mode (2 secs), an AGC target of 10e4 and an HCD collision energy of 30.

Raw files were searched using Proteome Discoverer software (v2.4, Thermo Scientific) using SEQUEST as search engine. We used both the SwissProt human database (updated

January 2020) and the Toxoplasma database (Release 44, ME49 proteome obtained from ToxoDB). The search for total proteome included variable modifications of methionine oxidation and N-terminal acetylation, and fixed modification of carbamidomethyl cysteine. Analysis of the phosphoproteome included carbamidomethylation on cysteine residues as a fixed modification, while phosphorylation on serine, threonine and tyrosine residues was set as variable modification. Trypsin was specified as the digestive enzyme. Mass tolerance was set to 10 pm for precursor ions and 0.2 Da for product ions. Peptide and protein false discovery rate was set to 1%.

Each analysis was performed with either two biological replicates (Co-IP) or three technical replicates (phosphoproteome). Before statistical analysis, peptide and phosphopeptide intensities were log2 transformed, normalized by the average value of each sample, and missing values imputed using a normal distribution 2 standard deviations lower than the mean. After data transformation, phosphopeptides were next normalized by protein abundance (i.e. normalized by cumulative protein intensities detected in $TiO_2$ flow through fractions) before assessing statistical changes in relative abundance. Statistical regulation in the phosphoproteomic datasets were assessed using a heteroscedastic T-test (if p-value < 0.05).

For the Co-IP data, individual peptide fold changes (TgPPM3C-HA vs. Control) for a given protein were calculated and averaged to obtain protein fold enrichment. P-values were then obtained from t-distributions and t-values calculated for each protein with at least two detected peptides by treating protein fold enrichment as the sample mean and using log transformed peptide intensity values to calculate the standard deviation, sample size, and degrees of freedom. The data distribution in both datasets were assumed to be normal but this was not formally tested. Fold-change cutoffs for both Co-IP and phosphoproteomic experiments were arbitrarily selected.

LC-MS/MS data from both Co-IP and phosphoproteomic experiments have been deposited onto the public repository Chorus under Project ID 1695 and deposited onto ToxoDB (EuPathDB).

## Hypergeometric testing

Hypergeometric testing for *Toxoplasma* compartment enrichment was performed with a custom R script (github.com/nataliesilmon/toxotools), in which the observed overlap between a set of genes (encoding either all detected phosphopeptides or only significantly altered phosphopeptides from ΔTgPPM3C samples) and a predefined set of genes belonging to an organelle compartment were compared to the random sampling of a similar number of genes from the entire *Toxoplasma* genome. The script was updated to include recent annotations and Gene IDs from ToxoDB (Release 47, ME49 genome). Generation of sequence logos was performed with Seq2Logo [57], using 31 amino acid sequence windows from either *Toxoplasma* phosphopeptides calculated as significantly enriched in ΔTgPPM3C samples or randomly selected *Toxoplasma* phosphopeptides detected in both wild type and ΔTgPPM3C samples.

## Mouse studies

Eight week old female C57Bl/6 mice (The Jackson Laboratory, Bar Harbor, ME) were infected with 16,000 tachyzoites of each strain intraperitoneally for survival analysis. Mortality was observed daily for 30 days. For cyst burden analysis, brains were collected from mice injected intraperitoneally with 2,000 parasites 30 days prior. One brain hemisphere per mouse was homogenized with a Wheaton Potter-Elvehjem Tissue Grinder with a 100–150 μm clearance (ThermoFisher) in PBS and an aliquot of the homogenate was viewed under an epifluorescence microscope (Nikon) to count GFP-positive cysts. Kruskal-Wallis tests and Dunn's

multiple comparisons test were performed to test for significance between groups with PRISM 8 software. A log-rank test was performed in PRISM to test for statistical significance in Kaplan-Meier survival curves.

## Supporting information

**S1 Fig. Host cell invasion, parasite replication, and parasite egress are unaffected in the ΔTgPPM3C strain. (A)** Parasite invasion was assessed by immunofluorescence 30 minutes post-addition of extracellular tachyzoites to an HFF monolayer, using antibodies to label parasites before and after permeabilization of host cells. Data from three independent experiments are shown. No significant differences were measured between the TgPPM3C-HA, ΔTgPPM3C, and TgPPM3C-COMP strains. **(B)** Parasite replication was assessed by immunofluorescence after 32 hours of infection, counting the number of parasites per vacuole from at least 150 vacuoles of each strain per experiment. Data from three independent experiments are shown. No significant differences were seen between the TgPPM3C-HA, ΔTgPPM3C, and TgPPM3C-COMP strains. **(C)** Parasite egress was assessed by immunoflourescence after adding either DMSO (control) or the calcium ionophore A23187 and quantifying the number of occupied (GRA6 positive, SAG1 positive) and unoccupied vacuoles (GRA6 positive, SAG1 negative) from at least 200 vacuoles per strain for each condition and experiment. Data from three independent experiments are shown. No significant differences were seen between the TgPPM3C-HA, ΔTgPPM3C, and TgPPM3C-COMP strains.
(TIF)

**S2 Fig. No gross differences in parasitophorous vacuole morphology are observed between TgPPM3C-HA and ΔTgPPM3C strains.** Representative transmission electron micrographs of vacuoles formed by either TgPPM3C-HA or ΔTgPPM3C parasites, 32 hours post-infection in human fibroblast monolayers. Tubules from the intravacuolar network, a hallmark of tachyzoite *Toxoplasma* vacuoles, can be seen in the vacuoles formed by both strains.
(TIF)

**S3 Fig. Endogenous GRA16 protein export is impaired and GRA16-MYR1 interaction is perturbed in the ΔTgPPM3C strain. (A)** Violin plots of endogenous GRA16-3xHA accumulation in host nuclei infected with either PruQ or ΔTgPPM3C parasites. GRA16 fluorescence was quantified from fibroblast nuclei containing a single parasite vacuole. A significant decrease in GRA16 host nuclear accumulation is observed during ΔTgPPM3C infection, suggesting defects in effector export from the parasitophorous vacuole. Data were collected from three independent experiments. Representative images from which effector intensity were quantified are shown on the right. Antibody to SAG1 was used as a parasite marker, DAPI as a host nucleus marker, and anti-HA antibody was used to detect GRA16-3xHA. Scale bar equals 10μm. **(B)** Immunoblots of protein lysates obtained from infected monolayers (left panel) or extracellular parasites (right panel). Infected monolayer lysates were treated (+AP) or mock treated (-AP) with 100U of alkaline phosphatase for 1hr in a 37C water bath. A shift in the larger of the two GRA16-3xHA bands (>55kDa) is observed following AP treatment of lysates obtained from both the GRA16-3xHA and ΔTgPPM3C strains, suggesting that this band represents phospho-GRA16. The larger GRA16-3xHA band is also detected in extracellular parasite protein lysates, indicating GRA16 is likely phosphorylated within parasites prior to secretion. Antibody to SAG1 or TgALD1 was used as a loading control. The infected monolayer lysate immunoblot with alkaline phosphatase treatment is representative of two independent experiments.
(TIF)

**S4 Fig. PPM3C protein alignment across coccidian relatives demonstrates high conservation in *Neospora caninum* and *Hammondia hammondi*, and across *T. gondii* strains.** Clustal Omega alignment of TgPPM3C amino acid sequences from the Type I reference strain GT1, Type II reference strain ME49, and Type III strain VEG. An amino acid alignment comparing select Apicomplexan PPM3C homologues is also shown for *Toxoplasma* strain ME49 (TGME49_270320), *Hammondia hommondi* strain H.H.34 (HHA_270320), *Neospora caninum* Liverpool (NCLIV_036340), *Sarcocystis neurona* SN3 (SN3_02500075), and *Eimeria acervuline* Houghton (EAH_00048430). Sequences were obtained from ToxoDB and EuPathDB. (TIF)

**S5 Fig. The X-ray crystal structure of TgPPM3C catalytic domain demonstrates a largely conserved tertiary structure. (A)** Ribbon projection of TgPPM3C crystal structure. The characteristic beta sandwich (red) formed by adjacent beta sheets can be seen coordinating a Praseodymium cation ($Pr^{3+}$) (white circle) in the putative active site of the enzyme. The side chains of the conserved metal and phosphate coordinating amino acids are shown, along with water molecules (small cyan circles) in this region. Alpha helices (blue) surround the beta sandwich. The flap domain (green) is seen above the beta-sandwich pocket, along with an additional domain (white) present beneath the flap domain, which is not seen in the human PPM1A crystal structure. **(B)** Top–surface mesh rendering of TgPPM3C, demonstrating the putative catalytic pocket in which the $Pr^{3+}$ ion is found. Coloring scheme is as described in **A.** Bottom–calculated Coulombic potential for the TgPPM3C surface mesh, demonstrating a strongly acidic (red) putative catalytic pocket with no basic residues (blue) near the beta-sandwich cleft. **(C)** Ribbon projection of the human PPM1A crystal structure. The beta sandwich (red) can be seen coordinating two $Mg^{2+}$ cations (purple circles) in the active site, along with a phosphate ion (orange stick figure) just above the $Mg^{2+}$ cations. The side chains of the conserved metal and phosphate coordinating amino acids are shown, along with water molecules (small cyan circles) in this region. The flap domain is shown in green, and the C-terminal domain extending away from the center of the enzyme is shown in light blue. **(D)** Top–surface mesh rendering of PPM1A, demonstrating a similar active site pocket as seen in TgPPM3C. Coloring scheme is as described in **C.** Bottom–calculated Coulombic potential for the PPM1A surface mesh, demonstrating the characteristic acidic (red) catalytic pocket and a few basic residues (blue) above the catalytic pocket. Molecular graphics and analyses of crystal structures were prepared using UCSF Chimera [59] and the publicly available structures of the TgPPM3C catalytic domain (Protein Data Bank ID: 2ISN) and human PPM1A (Protein Data Bank ID: 1A6Q). (TIF)

**S1 Table. List of oligonucleotides used for cloning and genetic manipulation.** (DOCX)

**S1 Data. LC-MS/MS data obtained from Co-IP experiments.** Data are presented in five different tabs. The "Results Summary Tab" lists the calculated fold enrichment (TgPPM3C-HA/Control) and -$\log_2$ p-values for all of the proteins detected in each replicate. The "Replicate 1/2 Analysis" tabs demonstrate data transformation steps and equations used to determine average protein fold enrichment and p-values. The "Replicate 1/2 Raw Data" tabs provide information on search engine identification quality parameters (e.g. PEP scores, Percolator q-values, etc.). This dataset has been deposited into the mass spectrometry open access repository Chorus under Project ID 1695. (XLSX)

**S2 Data. LC-MS/MS data obtained from phosphoproteome preps of wild type (TgPPM3C-HA) and ΔTgPPM3C cultures.** Data are presented in six different tabs. The "Ranked Parasite Phosphopeptide" tab summarizes the average fold change (ΔTgPPM3C/ Wild Type) and -log$_2$ p-values from a ranked list of all detected parasite phosphopeptides and also includes the specific phosphomodifications, short names, and whether a differentially regulated phosphopeptide belongs to a protein known to be secreted into the parasitophorous vacuole lumen. The "TiO$_2$ Flow Through Analysis", "TiO$_2$ Phosphopeptide Analysis", and "Fold Change Analysis" tabs demonstrate data transformation steps and equations used to determine average normalized phosphopeptide fold enrichments and p-values. Information on search engine identification quality parameters (e.g. PEP scores, Percolator q-values, etc.) are provided in the "TiO$_2$ Flow Through Raw Data" and "TiO$_2$ Enriched Peptide Raw Data" tabs. This dataset has been deposited into the mass spectrometry open access repository Chorus under Project ID 1695 and onto ToxoDB (EuPathDB).
(XLSX)

## Acknowledgments

We thank all members of the Weiss lab for their comments, suggestions, and insights in the preparation of this manuscript, with special thanks to Rama R. Yakubu and Jessica Weiselberg. We thank Dr. John Boothroyd and members of his lab for useful insights and experimental suggestions. We thank the Albert Einstein Analytical Imaging Facility, specifically Dr. Vera DesMarais and Andrea Briceno for assistance on various light microscopes, as well as Leslie Gunther-Cummins, Xheni Nishku, and Timothy Mendez for electron microscopy sample preparation and training. We thank the Einstein Laboratory for Macromolecular Analysis and Proteomics for their pivotal assistance with all LC-MS/MS preparations and analysis.

## Author Contributions

**Conceptualization:** Joshua Mayoral, Louis M. Weiss.

**Data curation:** Joshua Mayoral, Louis M. Weiss.

**Formal analysis:** Joshua Mayoral, Simone Sidoli, Louis M. Weiss.

**Funding acquisition:** Joshua Mayoral, Louis M. Weiss.

**Investigation:** Joshua Mayoral, Tadakimi Tomita, Vincent Tu, Jennifer T. Aguilan, Simone Sidoli, Louis M. Weiss.

**Methodology:** Joshua Mayoral, Tadakimi Tomita, Jennifer T. Aguilan, Simone Sidoli, Louis M. Weiss.

**Project administration:** Joshua Mayoral, Louis M. Weiss.

**Resources:** Joshua Mayoral, Tadakimi Tomita, Louis M. Weiss.

**Supervision:** Joshua Mayoral, Louis M. Weiss.

**Validation:** Joshua Mayoral, Louis M. Weiss.

**Visualization:** Joshua Mayoral, Louis M. Weiss.

**Writing – original draft:** Joshua Mayoral.

**Writing – review & editing:** Joshua Mayoral, Simone Sidoli, Louis M. Weiss.

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
