## [Decision Letter · Decision Letter 0]

3 Aug 2020

Dear Dr. Weiss,

Thank you very much for submitting your manuscript "Toxoplasma gondii PPM3C, a secreted protein phosphatase, affects parasitophorous vacuole effector export" for consideration at PLOS Pathogens. As with all papers reviewed by the journal, your manuscript was reviewed by members of the editorial board and by several independent reviewers. In light of the reviews (below this email), we would like to invite the resubmission of a significantly-revised version that takes into account the reviewers' comments.

The reviewers had mixed opinions regarding the paper although I believe that this represents a significant advance for the field. But, the reviewers raise important concerns/points. While all of their comments should be addressed, particular attention should be paid to: providing higher quality immunofluorescence images, defining which lytic growth step is affected by loss of PPM3c, examining, if possible, the impact of PPM3c on Myr1 phosphorylation and interactions, addressing why the phosphomimetic and phosphoablative GRA16 mutants have similar phenotypes, and better integrating this work with that of Cygan et al.

We cannot make any decision about publication until we have seen the revised manuscript and your response to the reviewers' comments. Your revised manuscript is also likely to be sent to reviewers for further evaluation.

Sincerely,

Ira J Blader

Guest Editor

PLOS Pathogens

Kirk Deitsch

Section Editor

PLOS Pathogens

Kasturi Haldar

Editor-in-Chief

PLOS Pathogens

orcid.org/0000-0001-5065-158X

Michael Malim

Editor-in-Chief

PLOS Pathogens

orcid.org/0000-0002-7699-2064

The reviewers has mixed opinions regarding the paper although I believe that this represents a significant advance for the field. But, the reviewers raise important concerns/points. While all of their comments should be addressed, particular attention should be paid to: providing higher quality immunofluorescence images, defining which lytic growth step is affected by loss of PPM3c, examining, if possible, the impact of PPM3c on Myr1 phosphorylation and interactions, addressing why the phosphomimetic and phosphoablative GRA16 mutants have similar phenotypes, and better integrating this work with that of Cygan et al.

Reviewer's Responses to Questions

**Part I - Summary**

Reviewer #1: Mayoral et al investigate the role of the secreted phosphatase PPM3C during Toxoplasma infection. They show this PV/cyst-localised phosphatase has a critical role in acute infection and a modest growth defect in vitro. Using co-IP and phosphoproteomics to identify interaction partners/targets they identify both translocon component and known secreted proteins. PPM3C dependent defects in secretion are subsequently shown for GRA16 and GRA28, although importantly not for IRF1 phenotype that is dependent on TgIST. Phosphomutants of GRA16 further indicate a role of phosphoregulation in the secretion of this protein. Overall, I think this is an interesting piece of work and a valuable contribution to the field. While it was shown in 2011 that many secreted proteins are phosphorylated, further developments in our understanding have been limited and the critical role of PPM3C during infection and involvement in protein secretion into the host cell is noteworthy. However, I was disappointed that the MYR1 phosphorylation was not investigated further as insight into regulation of the translocon could be more insightful than individual effectors. Additionally, PPM3C localisation has been previously reported and its contribution to effector translocation investigated yet the authors did not discuss these conflicting results.

Reviewer #2: Toxoplasma can export GRA effectors beyond the parasitophorous vacuole membrane. These GRA effectors play important roles in the modulation of host cell signaling pathways and the host immune response. Therefore identifying the exact mechanisms of GRA translocation are important. Multiple Toxoplasma proteins have now been identified that affect GRA effector translocation, some of these are part of the translocon while others seem to affect the correct folding or phosphorylation status of either translocon components or of the secreted effectors. Here Mayoral et al. show that the PPM3C secreted protein phosphatase affects the export of some GRA effectors likely by affecting their phosphorylation status. They used multiple independent methods to show that it is likely that PPM3C affects the export of GRA effectors by modulating their phosphorylation status. Overall the experiments were well performed and the conclusions valid. This manuscript will be of interest to the field.

Reviewer #3: The goal of this paper is to follow up on a putative phosphatase (TgPPM3C) identified from a prior proteome study of cyst wall. Here the authors seek to show the effect of TgPPM3C on the phosphostate of Toxoplasma proteins, and how this impacts 2 dense granule proteins that are exported via the MYR1-complex. The strength of the paper is the concept that phosphostate (via TgPPM3C) might affect dense granule protein secretion through the MYR-complex. Unfortunately, the data are not convincing and contradict a recently published paper (ref 16) that used more rigorous methods.

**Part II – Major Issues: Key Experiments Required for Acceptance**

Reviewer #1: Novelty: The authors don’t mention that PV localisation and interaction with MYR1 have been previously reported for this protein (despite referencing the paper to say it supports their c-myc experiment). This previous paper uses a different strain of Toxoplasma (RH) but shows that PPM3C KO has no effect on parasite growth, GRA16 nuclear translocation or cMyc upregulation. While there could be strain differences, I think it is essential that the authors discuss rather than ignore these previous results.

Fig 2. In these initial experiments the strains TgPPM3C-HA, deltaTgPPM3C, and TgPPM3C-COMP are used whereas later Pruku80 is used with deltaTgPPM3C. Is there any difference between TgPPM3C-HA and Pruku80 in the plaque assay or GRA16 translocation/c-myc experiments?

Did the authors look in to the MYR1 phoshorylation sites ie whether phosphomutants impact effector translocation? Given this is the only protein identified in both the co-IP and phosphoproteome, this seems central to the story to follow up.

Alternatively, as an interaction between MYR1 and GRA16 has previously been shown, have the authors investigated whether GRA16 phosphomutants impacts this?

Reviewer #2: 1) The PPM3C parasites seem to grow slower based on the plaque assays. For protein export were vacuoles with similar number of parasites/vacuole elected for nuclear quantification of exported proteins? It seems that the reduced growth could affect this quantification. It appears that the authors were careful to select only host cells with a single vacuole. Maybe they can go back to the images and at least investigate if the parasites/vacuole influenced these values.

2) It was a bit unclear how the statistical analyses on the quantifications of the IFAs (and plaque areas) performed by the authors were performed. The authors state that the number of observations made for quantification are provided above each violin plot and Kruskal-Wallis tests were performed to calculate p-values. From the material and methods it was unclear if they did the statistical analysis on the three means from the three independent biological experiment or if they used all the data from all the individual cells. Individual host cells are not independent observations in a single biological experiment. It seems more correct to use the average of each biological experiment as one observation and perform the statistics on the 3 averages from the 3 independent biological experiments (but maybe this is what they did?). Can the 3 independent averages be indicated in the violin plots?

3) It was unclear why the phosphomimetic and phosphoablative mutations in GRA16 both showed reduced nuclear export. Maybe the authors could perform these experiment in the knockout and complemented PPM3C strain and show that in the knockout there is no difference in the amount localized to the host nucleus.

Reviewer #3: 1. Fig 1B, C, D- the staining is very poor quality and the single images are concerning for the HA tag having screwed up the protein’s localization. In particular, C,D do not look like any typical pattern of staining I have seen for proteins secreted into the vacuole. The authors will need to show more convincing data that this protein is in dense granules/secreted into the PV (could the it be ending up in the residual body?) (see ref 16 Fig2B for more typical staining and Fig S2 for staining in the parasite)

2. Fig 2B- since there a reasonably impressive in vitro defect, the authors should determine if the defect in growth is at the level of attachment/invasion, replication, or egress, especially as in Fig 3, the authors point out that knocking out TgPPM3C affects the phosphorylation state of proteins in organelles involved in most of those processes. (FYI- phenotype different than what was seen in ref 16)

3. Fig 4A,B- This paper finds a mild phenotype with PPM3C KO but ref 16 found no phenotype on Gra16 translocation. These experiments would need to be repeated with a stable GRA16-HA line (akin to ref 16) to confirm these differences because the MYR-machinery/phenotypes are predicted to be conserved between RH (ref 16) and Pru (current paper)

4. Fig 4C- to show that TgIST is not affected by the loss of TgPPM3C, they need to do the same type of experiments as in Fig A (transient expression of an epitope-tagged TgIST)

5. Fig 5- Re-do the experiments with a glutamine substitution which most closely mirror the phosphomimetic glutamate substitution. Figure out a way to show that these substitutions don’t just “break” GRA16. The current data could be interpreted as both glutamate and proline disrupt GRA16’s export (or the protein itself) in a way that has nothing to do with the phosphostate.

**Part III – Minor Issues: Editorial and Data Presentation Modifications**

Reviewer #1: Fig1. Please add comment that PV localisation has been shown. Additionally, do you have any hypotheses as to why this looks different? Do the authors ever see signal throughout the vacuole rather than the aggregation you show? Whilst MYR1 is not always PVM localised, how would they envisage this aggregation of PPM3C you show fits with dephosphorylating MYR1 and secreted proteins?

From the Fig. 2D images the larger plaques present in TgPPM3C-HA but not deltaTgPPM3C look like merged neighbouring plaques. How do they account for this in the analysis? Do they have corresponding data on the number of plaques per strain (looks reduced)? As before please add comment on the difference to RH data. Do the authors think this is a true strain difference or due to experimental differences?

Table 1. The co-IP experiment gave very few interacting partners. Whilst the short-term interactions of the enzyme could explain this I was surprised by the use of sonication in the protocol. Is this routine in their lab? I thought this high energy treatment disrupted many protein:protein interactions (even if an ice bath is used)?

Also, please add comment that MYR1-PPM3C interaction was previously reported.

It would be good to see TgIST nuclear localisation data in the PPM3C KO (currently data not shown) with only the IRF1 phenotype shown in the Figure. Could there be slightly reduced TgIST secretion and still observe the IRF1 as analysed here? (Ie perhaps surplus is secreted)

Fig 4 GRA16 and GRA28 nuclear translocation: The PPM3C KO looks like single parasite vacuoles with the SAG1 staining in contrast to a 4 pack with the PruQ. Was there a difference in growth rate with the PruQ – singles/double parasites seem very low for a 24h timepoint. Have you tried normalising the nuclear signal to that in the vacuole to help account for these differences? Otherwise could the reduced nuclear level not reflect the reduced protein in the vac? And again, the contrasting phenotype to the published RH PPM3C KO needs discussed.

Minor:

I found that both the introduction and discussion were overly focussed on chronic infection and lacked other key information while the data addressed acute infection – for example writing more regarding cyst wall components rather than commenting secreted proteins are extensively phosphorylated (after secretion). Whilst I understand the cyst wall proteome led them to this phosphatase, this could be explained in a sentence. Additionally, chronic kinases are mentioned with more emphasis than the PV localised WNG1 that impacts membrane structures in the PV and is implicated in the phosphorylation of some of the targets identified.

Lines 128-132, Fig 2C/D and legend

The figure legend and methods are not consistent on the inoculum for the cyst counting experiment. Please check – and having the number in the main text would be more helpful than ‘equal inoculums’, especially as the information is there for the survival experiment.

Figure S1 The IVN looks normal in the PPM3C KO as stated but the image for TgPPM3C-HA does not seem to have clear tubules. Is this just the image or were there any differences between this and the parental line?

Line 173 Consider rewording ‘processed through LC-MS/MS to obtain peptide spectra’ to ‘analysed by LC-MS/MS’ or something similar.

Line 183 ‘more than half (79) of all differentially abundant phosphopeptides belong to proteins previously known to be secreted into the lumen of the parasitophorous vacuole (Dataset S2).’

Can this information be added to the table?

Line 206:’ In line with our initial hypothesis’

It would be helpful to remind the reader what this is.

Fig 3D/methods – it would be helpful if you added a comment on what hypergeometric testing is -this phrase is used several times but with no description even in the methods.

Line 212: This logic seems a bit flawed as intravacuolar components also show virulence defects in vivo so there doesn’t seem to be a reason why PPM3C targets leaving the vacuole would be more important than intravacuolar.

Line 237: I find this reasoning a little narrow – PPM3C modification of MYR1 could still have distinct impacts on different subsets of effectors. We understand so little about this mechanism, perhaps MYR1 phosphorylation affects its interactions with other translocation components, chaperones or effectors.

Line 255: th to the

Fig 4A/B: Please add that this is 24h infection to the legend.

Fig 4A: It is stated that you only analyse cells with one PV but the image appears to have two?

Fig 4B: Is the myc staining panel only myc signal currently? I haven’t seen the signal from the parasites before.

Figure 4C: Please add uninfected and PRuQ labels to the images (as the rest are labelled).

Line 484: Please add magnification used for these fields of view.

Fig 6: This is perhaps more suited as a supplementary figure and I think needs less attention in the discussion.

Dataset S1

Please check column names eg ‘PruQ (ctrl) Tac’ only doesn’t describe what the number actually refers to.

Also, I found the format of empty columns with a title referring to following columns initially unclear (looks like missing data). I would suggest changing these to the row above (personal opinion only).

Combined analysis tab – Rep 1 p values showing as div/0 error

Is the ‘Reference cells from Rep2 sheet for indexing’ meant to be there?

Combined analysis results tab – p values are >10. I assume this is -log p, please label as such.

Methods: Most of this is written short format so that it could not be easily repeated.

Dataset S2

Remove *Absent in all control samples (candidate for mutagenesis?) from row 663

There is no legend – a description of the tabs would be helpful as its quite unclear, especially which to focus on of the normalised, cumul_normalised volcano and the final corrected tab.

Data availability:

Acquired RAW files should be deposited in a dedicated mass spec open access repository, like ProteomeXchange via PRIDE, for example.

Co-IP experiment:

1) The authors claim they did this experiment in 2 independent replicates (N=2). Could they explain how they managed to obtain individual p-values for replicate 1 and 2 (S1 Dataset) where they have no technical replicates just the co-IP and control?

2) How was the reported fold change cut off estimated in this experiment?

Note: in the manuscript text it is reported as FC > 1.5 whereas in the S1 Dataset it says FC > 1.

3) No search engine generated identification quality parameters, such as identification scores, number of unique peptides per protein or PEPs are given in the S1 dataset.

Phosphoproteome experiment:

1) How was the fold change (-/+ 1.2) cut off selected in this experiment?

Note: in the manuscript text it is reported as FC -/+ 1.2 whereas in the S2 Dataset it says FC -/+ 1.5 (Normalized results tab)

2) The number of detected phosphopeptides is actually 5,881 (total) and 2,387 (Toxo) as per S2 Dataset/tab Phospho.

3) S2 Dataset needs some clarification:

a) No search engine generated identification and phosphosite localization parameters are reported. One can only assume that the value in parenthesis by the site info could be localization probability?

b) In several instances within different tabs (Normalized results and Norm Volcano tabs) the authors seem to use normalized phosphosite intensities (per sample) and then report the FC for a phosphopeptide. This does not seem to be correct taking into account the manuscript text, Fig 3A and S2 Dataset where authors report they performed quants and normalization on the phosphopeptide level. Please clarify.

c) It would be easier for a reader to find their way around the table if the host and parasite entries were separated into different tabs. The authors could also provide the list of significantly changing Toxoplasma phosphopeptides/phosphoproteins in a separate tab since this is the most important piece of information the reader looks for when browsing that table.

d) What is the blue colour coding in Normalized results and Volcano tabs?

Reviewer #2: 1) It was difficult to understand the S1 dataset. In the combined analysis datasheet why does each phosphopeptide have the same P-value and fold-change? When a phosphopeptide was not detected in one strain but it was in the other it seems a value was imputed that was 2 SD below the mean. Is the rationale for this because this is the limit of detection? Why not use the value of the lowest detectable peptide?

2) Fig. 4a states that the data were collected from 3 technical replicates. Therefore, it appears this experiment was only performed once. The authors should repeat the experiment and present averages from at least 3 independent experiments.

Reviewer #3: Fig 3C,D- proteins with increase phosphopeptides in the TgPPM3C KO do not need to be direct substrates of TgPPM3C. Thus, not finding a consensus sequence within them, does not rule out that TgPPM3C has a consensus sequence.

PLOS authors have the option to publish the peer review history of their article (what does this mean?). If published, this will include your full peer review and any attached files.

Reviewer #1: No

Reviewer #2: No

Reviewer #3: No
---

## [Decision Letter · Decision Letter 1]

16 Nov 2020

Dear Dr. Weiss,

Thank you very much for submitting your manuscript "Toxoplasma gondii PPM3C, a secreted protein phosphatase, affects parasitophorous vacuole effector export" for consideration at PLOS Pathogens. As with all papers reviewed by the journal, your manuscript was reviewed by members of the editorial board and by several independent reviewers. The reviewers appreciated the attention to an important topic. Based on the reviews, we are likely to accept this manuscript for publication, providing that you modify the manuscript according to the review recommendations.

As you will read, the reviewers were in general agreement that you addressed most of their concerns. Reviewer #3 was, however, still concerned about the serine to proline mutants. While I share some of these concerns, I feel that since the mutations lie in a disorganized domain of the protein it is difficult to predict the impact of these substitutions. But addressing this would require a substantial level of structure/function assays that, due to COVID, I am afraid would take an overly significant amount of time to complete. I do, however, agree with the reviewer's suggestion about showing images of and quantifying uninfected nuclei.

Sincerely,

Ira J Blader

Guest Editor

PLOS Pathogens

Kirk Deitsch

Section Editor

PLOS Pathogens

Kasturi Haldar

Editor-in-Chief

PLOS Pathogens

orcid.org/0000-0001-5065-158X

Michael Malim

Editor-in-Chief

PLOS Pathogens

orcid.org/0000-0002-7699-2064

As you will read, the reviewers were in general agreement that you addressed most of their concerns. Reviewer #3 was, however, still concerned about the serine to proline mutants. While I share some of these concerns, I feel that since the mutations lies in a disorganized domain of the protein it is difficult to predict the impact of these substitutions. But addressing this would require a substantial level of structure/function assays that, due to COVID, I am afraid would take an overly significant amount of time to complete. I do, however, agree with the reviewer's suggestion about showing images of and quantifying uninfected nuclei.

Reviewer Comments (if any, and for reference):

Reviewer's Responses to Questions

**Part I - Summary**

Reviewer #1: In the revised version the authors have adressed my concerns and I think this is a very interesting contribution to our understanding of the development of the intracellular niche of Toxoplasma in a host cell. The role of differential phosphorylation of secreted proteins remains an enigma and the role of phosphatases has previously not been adressed. Of particular interest is the specific effect of an export defect of only some, but not all proteins that are exported further into the host cell. This argues for a complex pattern of protein regulation of secreted proteins.

Reviewer #2: The authors have addressed all my comments.

Reviewer #3: The authors have addressed my concerns in terms of improving the staining for TgPPM3C; addressing invasion, replication, and egress; and performing their assays with a clonal, endogenously tagged GRA16 and a transiently expressing epitope tagged TgIST. Unfortunately, they chose to not address the concerns about using a proline substitute for the phospoablative GRA16. Basically- if your positive control doesn’t act as a positive control, that is a problem. This issue calls into question the results of the phosphomimetic, which is the lynchpin to the novelty and impact of this work, especially given the Cyagen study which has significantly different findings (which may or may not relate to be strain specific differences).

**Part II – Major Issues: Key Experiments Required for Acceptance**

Reviewer #1: No major issues.

Reviewer #2: None.

Reviewer #3: You will need to re-do the phosphoablative studies with a GRA16 uses alanine or glutamine substitutions for serine.

**Part III – Minor Issues: Editorial and Data Presentation Modifications**

Reviewer #1: I am happy with the current version. It may be worthwhile to discuss these results in the light of a recent publication (Young et al., 2020 mSphere) that describes differential phosphorylation between the Tachyzoite and Bradyzoite stage- and if proteins that are targets of PPM3C are found among these, maybe implying a wider role for this phosphatase in latent stage parasites.

The one experiment I would love to see - although not a request for publication of this manuscript as slightly beyond the scope - whether PPM3C KOs have a defect in cystwall formation in vitro. This could illuminate whether PPM3C has an additional defect in cyst formation.

Reviewer #2: The differences in Fig. 4B and 5B don't seem very large and the volcano plots largely overlap between the different groups. It seems surprising that these are so highly significant. Maybe the authors can double check their analyses.

Reviewer #3: Fig 2 B: please quantify your plaque assays as well (i.e. not just plaque size but also number)

Fig 4A: Gra24- which they are using as a negative control, for both PruQ and delta TgPPM3C, has an intensity at the level of uninfected cells (see Fig 4B) or cells in which the Gra protein is not secreted (intensity level for Gra28 in TgPPM3C parasites). This suggests they are not detecting Gra24 secretion in PruQ or TgPPM3C, which is consistent with the provided IFA. To address this issue, the authors should provide an image with nuclei from infected and uninfected host cells (akin to 4B COMP image, 4C, and 4D).

Line 290 – 297: The results with c-Myc are more consistent with ref 25 rather than ref 21

Line 312: For improved accuracy, the phrase “GRA16 expressed ectopically” should be revised to “GRA16 transiently ectopically expressed” (the images in 4A were generated by transient expression as far as I can tell)

Line 317 – 330/Fig S3B: How many phosphate groups are being added to GRA16? As each phosphate group adds ~80 Da, so the bands in S3B don’t seem quite right. (The lower band in the gel from the extracellular parasites seems reasonable, but the upper band seems extreme)

Line 356 – 369/Fig 5C: I’m not sure how these experiments show that the mutations in amino acid content in GRA16 don’t affect protein stability.

PLOS authors have the option to publish the peer review history of their article (what does this mean?). If published, this will include your full peer review and any attached files.

Reviewer #1: No

Reviewer #2: No

Reviewer #3: No
---

## [Editor Report · Decision Letter 2]

23 Nov 2020

Dear Dr. Weiss,

We are pleased to inform you that your manuscript 'Toxoplasma gondii PPM3C, a secreted protein phosphatase, affects parasitophorous vacuole effector export' has been provisionally accepted for publication in PLOS Pathogens.

Best regards,

Ira J Blader

Guest Editor

PLOS Pathogens

Kirk Deitsch

Section Editor

PLOS Pathogens

Kasturi Haldar

Editor-in-Chief

PLOS Pathogens

orcid.org/0000-0001-5065-158X

Michael Malim

Editor-in-Chief

PLOS Pathogens

orcid.org/0000-0002-7699-2064
---

## [Editor Report · Acceptance letter]

16 Dec 2020

Dear Dr. Weiss,

We are delighted to inform you that your manuscript, "Toxoplasma gondii PPM3C, a secreted protein phosphatase, affects parasitophorous vacuole effector export," has been formally accepted for publication in PLOS Pathogens.

Best regards,

Kasturi Haldar

Editor-in-Chief

PLOS Pathogens

orcid.org/0000-0001-5065-158X

Michael Malim

Editor-in-Chief

PLOS Pathogens

orcid.org/0000-0002-7699-2064